# Multi-annual modes in the 20th century temperature variability in reanalyses and CMIP5 models

Heikki Järvinen[1], Teija Seitola[1,2], Johan Silén[2], and Jouni Räisänen[1]

[1]Department of Physics, University of Helsinki, Finland
[2]Finnish Meteorological Institute, Helsinki, Finland

*Correspondence to:* Heikki Järvinen (heikki.j.jarvinen@helsinki.fi)

**Abstract.** A performance expectation is that Earth system models simulate well the climate mean state and the climate variability. To test this expectation, we decompose two 20th century reanalysis data sets and 12 CMIP5 model simulations for years 1901 – 2005 of the monthly mean near-surface air temperature using Randomised Multi-Channel Singular Spectrum Analysis (RMSSA). Due to the relatively short time span, we concentrate on the representation of multi-annual variability which the RMSSA method effectively captures as separate and mutually orthogonal spatio-temporal components. This decomposition is a unique way to separate statistically significant quasi-periodic oscillations from one another in high-dimensional data sets.

The main results are as follows. First, the total spectra for the two reanalysis data sets are remarkably similar in all time scales, except that the spectral power in ERA-20C is systematically slightly higher than in 20CR. Apart from the slow components related to multi-decadal periodicities, ENSO oscillations with approximately 3.5 yr and 5 yr periods are the most prominent forms of variability in both reanalyses. In 20CR, these are relatively slightly more pronounced than in ERA-20C. Since about the 1970's, the amplitudes of the 3.5 yr and 5 yr oscillations have increased, presumably due to some combination of forced climate change, intrinsic low-frequency climate variability, or change in global observing network. Second, none of the 12 coupled climate models closely reproduce all aspects of the reanalysis spectra, although some models represent many aspects well. For instance, the GFDL-ESM2M model has two nicely separated ENSO periods although they are relatively too prominent as compared with the reanalyses. There is an extensive Supplement and Youtube videos to illustrate the multi-annual variability of the data sets.

*Keywords:* spatio-temporal modes, climate variability, climate model simulation, random projection, RMSSA algorithm, ENSO oscillation, Youtube video

## 1   Introduction

The ultimate goal in developing Earth system models (ESM) is to enable exploitation of the inherent Earth system predictability, and hence reduce weather and climate related uncertainties in our daily life, and guide societies in making sustainable choices (e.g., Slingo and Palmer 2011; Meehl et al. 2014). For the predictions to be useful and usable, the expectation is that the climate mean state and climate variability are well simulated by these tools. Due to the complexity of the models and the data they

produce, testing the expectation poses a challenge: many aspects of the model performance are gathered under the variability concept and no single diagnostic alone is sufficient to exhaust its all facets. Yet, understanding the discrepancies between the observed and simulated variability is crucial feedback for model development.

Representation of climate variability among models participating in climate model inter-comparisons, such as CMIP5, has been studied by e.g. Bellenger et al. (2014), Knutson et al. (2013), Ba et al. (2014), and Fredriksen and Rypdal (2016). We will add to this literature by interfacing a representative set of contemporary coupled climate models with reanalysis data focusing on spatio-temporal modes of climate variability. One century covered with global reanalysis data is naturally very short for this purpose and severely constrains inter-comparison studies (e. g. Wittenberg 2009 and Stevenson et al. 2010). First, time series should cover a sufficient number of recurring "events" for obtaining significance for the findings. Therefore, decadal-to-multi-decadal variability is of interest but not as informative as focusing on shorter cycles of variability. Second, the applied methods have to be very effective in extracting information from the short but high-dimensional data sets. For these reasons, we concentrate on the representation of multi-annual variability in reanalyses and coupled climate models applying Randomised Multi-Channel Singular Spectrum Analysis (RMSSA; Seitola et al. 2014, 2015) which effectively separates mutually orthogonal spatio-temporal components from our high-dimensional data sets.

The aim of this study is to decompose the 20th century climate variability into its multi-annual modes, and to assess how these modes are represented by the contemporary climate models. We hope this to provide guidance for model development due better understanding of the deficiencies in representing reanalysed modes of multi-annual climate variability. Ultimately, interpreting the hints about model deficiencies as development topics are due for the development teams themselves. Our role is to point towards the potential error sources. For reassuring the teams that high-dimensional time series analysis is possible today, we emphasise the methodological aspect of this study. RMSSA can, under very weak assumptions on the data, decompose high-dimensional data sets in a unique way and separate statistically significant quasi-periodic spatio-temporal oscillations from one another. This is in contrast to many other approaches which either make assumptions about the oscillation structures, such as Fourier or spherical decomposition, or resolve only either spatial or temporal aspects of variability. RMSSA can detect spatially evolving "chains of events" through resolving eigenmodes of spatio-temporal covariance data. This is a significant advantage, say, over PCA which only resolves eigenmodes of spatial covariances and often projects temporal evolution of an "event" onto a number of different eigenmodes. In addition, the novel data compression based on random projections enable here a vast increase in tractable problem size (i.e., data dimension) - even multi-variate decomposition is now possible, although not included here.

## 2 Methods and Data

### 2.1 Randomised multi-channel singular spectrum analysis

Multi-channel singular spectrum analysis (MSSA; Broomhead and King, 1986a,b) can be characterised as being a time series analysis method for high-dimensional problems. It effectively identifies spatially and temporally coherent patterns of a data set by decomposing a lag-covariance data matrix into its eigenvectors and eigenvalues (e.g., Ghil et al., 2002) using singular value

decomposition (SVD). The lag window in MSSA is a user choice, recommended typically to be shorter than approximately one third of the length of the time series (Vautard and Ghil, 1989). Long lag window enhances the spectral resolution, i.e., the number of frequencies that can be identified, but distributes the variance on a larger set of components. MSSA eigenvectors are called here space-time EOFs (ST-EOFs), and the projections of the data set onto those ST-EOFs space-time principal

components (ST-PCs). Because of the lag window, ST-PCs have a reduced length and they cannot be located into the same index space with the original time series. However, they can be represented in the original coordinate system by the reconstructed components (RC; Plaut and Vautard, 1994).

MSSA is computationally expensive and practical limits are easily exceeded for large data sets and long lag windows. In order to overcome this limitation, a computationally more efficient variant, called Randomised MSSA (RMSSA; Seitola et

al., 2015), is applied here. The RMSSA algorithm, in a nutshell, (1) reduces the dimension of the original data set by using so-called random projections (RP; Bingham and Mannila, 2001; Achlioptas, 2003), (2) decomposes the data set by calculating standard MSSA in the low-dimensional space, and (3) reconstructs the components in the original high-dimensional space.

In RP, the original data set is projected onto a matrix of Gaussian distributed random numbers (zero mean and unit variance) in order to construct a lower dimensional representation. In this study, we reduce the data volume to about 5 % of the original

volume. Since the computational complexity of RP is low, involving only a matrix multiplication, it can be applied to very high-dimensional data sets. Although RP is not a lossless compression, it has the important property that the lower-dimensional data set has essentially the same structure as the original high-dimensional data set. This has been demonstrated for climate model data in Seitola et al. (2014). The RMSSA algorithm is briefly presented in the Appendix A.

## 2.2   Computation of spectra

The ST-PCs represent the different oscillatory modes extracted from the data set. In order to estimate the dominant frequencies associated with each ST-PC, the power spectrum is calculated with the Multitaper spectral analysis method (MTM) (Thomson, 1982; Mann and Lees, 1996). To further compare the variability modes and their intensities in different data sets, the power spectrum of all the ST-PCs of each data set is summed up to obtain so-called total spectrum. The ST-PCs are already weighted by their respective explanatory power, i.e. multiplied by the corresponding eigenvalue. Therefore the components with more

explanatory power have also higher spectral densities compared to the ones that explain only a small fraction of the variance. Therefore no extra weighting is needed in this step.

The uncertainty related to the explanatory power of each ST-PC (i.e. the confidence interval of the respective eigenvalue) is estimated using the Norths rule of thumb for sampling errors (North et al., 1982). The sampling error $(e_k)$ is given by $e_k \sim \lambda_k(2/N)$, where $\lambda_k$ is the eigenvalue associated with the $k^{th}$ ST-PC and $N$ is the length of the time series. Thus, the

confidence interval of the total spectrum describes the uncertainties related to the explanatory power of each ST-PC.

## 2.3   Statistical significance testing

In data sets of dynamical systems, ST-PCs/ST-EOFs of MSSA often appear as quadratic pairs that explain approximately the same variance and are $\pi/2$ out of phase with each other. However, existence of such a pair does not guarantee any physical

oscillation in the data set, and it may be due to some non-oscillatory processes, such as first-order autoregressive noise. Allen and Robertson (1996) formulated a test, where the oscillatory modes identified with MSSA are tested against a red noise null-hypothesis through Monte Carlo simulation.

Significance testing in MSSA requires solving conventional PCs of the original data set. In case of very high-dimensional problems this easily exceeds practical computational limits. The RMSSA implementation in Seitola et al. (2015) contains the Allen-Robertson test such that the PCs are solved in the dimension-reduced space, and is thus affordable even in very high-dimensional problems. The Appendix A also includes a short description of the significance test.

## 2.4 Data sources

The data consists of the monthly mean near-surface air temperature from the historical 20th Century simulations of 12 different climate models (Table 1). The selected models originate from different modelling centres, and thus do not have close common ancestor models. Furthermore, the selected models have undergone a long (generally several generations of) history of development, suggesting that the chosen models collectively represent the state-of-the-art. Near-surface temperature was chosen, because many processes must be adequately represented in coupled models to realistically capture the observed temperature distribution (Flato et al., 2013). These include processes in the Earth system component models (atmosphere, ocean, etc.) as well as in their mutual coupling models. Also, for the near-surface temperature, there are corresponding reanalysis data available.

The historical (1901–2005) simulations were extracted from the CMIP5 data archive and they follow the CMIP5 experimental protocol (Taylor et al. 2012). The 20th Century simulations use the historical record of climate forcing factors such as greenhouse gases, aerosols, solar variability, and volcanic eruptions. We used a single ensemble member of each model and the model data sets were interpolated into a common grid of $144 \times 73$ points.

As a reference, we used two reanalysis data sets: the 20th Century Reanalysis V2 data (hereafer 20CR) provided by the NOAA/OAR/ESRL PSD (Compo et al., 2011), and ERA-20C data provided by ECMWF (Poli et al., 2013). The data sets are produced using an ensemble of perturbed reanalyses, and the final data set corresponds to the ensemble mean. In 20CR, only surface pressure observations are assimilated, and the observed monthly sea-surface temperature and sea-ice distributions from HadISST1.1 (Rayner et al., 2003) are used as boundary conditions (Compo et al., 2011). In ERA-20C, observations of surface pressure and surface marine winds are assimilated (Poli et al., 2013). Unlike 20CR, it uses a more recent sea-surface temperature and sea ice cover analysis from HadISST2 (Rayner et al., 2006). Both 20CR and ERA-20C are forced by historical record of changes in climate forcing factors (greenhouse gases, volcanic aerosols and solar variations). In order to be consistent with the climate model simulations, the same time period is used (1901–2005, i.e., 1260 monthly mean fields) and the reanalysis data sets were interpolated into the same grid as the model simulations ($144 \times 73$ points).

## 2.5 Data processing

Some pre-processing of the data was needed before applying RMSSA. At each grid point the data sets were processed as follows:

- linear trend was fitted and removed,

- annual cycle was estimated using Seasonal-Trend Decomposition (STL; Cleveland et al., 1990) and removed,

- resulting values were mean-centered and divided by the average standard deviation of all the data sets (see Figure 1). Average standard deviation was obtained after removal of the trend and the annual cycle.

5      The reanalysis and climate model data sets have different temperature standard deviations, which would impact the temperature variability from inter-annual to multi-decadal timescales (e.g., Thompson et al. 2015). To retain these differences, we have used a common normalization factor (i.e., the average standard deviation of all the data sets). This procedure reduces the weight of grid points with high variance, typically at higher latitudes, and hence adds weight on the lower latitude features. After the pre-processing, the dimension reduction step of RMSSA was applied so that approximately 5 % of the original data 10   dimensions were retained. The lag window in the analysis was 20 yr (240 months). The total spectra were obtained from this analysis, and are comparable due to normalisation using the common standard deviation of the data sets.

     The statistical significance test uses a red noise null hypothesis. In the test we have used data sets that are normalised by their own standard deviations. Using a common normalisation interferes with generating the red noise surrogates corresponding to each data set. The first 50 PCs of each data set were retained as input. Those PCs explain 79 % of the variability in 20CR, 75 15   % in ERA-20C, and 70 %–80 % in the climate model data sets. A total of 1000 realisations of red noise surrogate data sets were generated, and confidence interval (95 %) for the oscillatory modes were estimated. We note that transformation to PCs may interfere with the detection of weak signals, as demonstrated by Groth and Ghil (2015).

### 2.6   Data visualization

     We used reconstructed components (RC; see Appendix A) for visualisation of the spatial patterns related to ST-PCs. For each 20   grid point time series, we can calculate the RCs corresponding to the ST-PCs (or modes) of interest. These RC values, reflecting the contribution of each grid point to the mode, can be plotted on a map at each time step. We have used these maps to construct videos of the spatio-temporal modes. In Section 3.5, we have analysed RCs corresponding to 3–4 yr variability. The result is a time series of the data corresponding to the 3–4 yr mode in each grid point and according to its variance after detrending and removing the annual cycle. In the analysis we have neglected 5 yrs in the beginning and the end of the time series, because 25   the reconstruction procedure may be biased there (see the Appendix, eq. A4). The videos can be found at our Youtube channel (https://www.youtube.com/channel/UCu1zJdwJfLaXvfvTqsKCLHw).

     To summarise the animations, we have calculated composite maps of the modes. The compositing procedure follows the one described in Plaut and Vautard (1994). The idea is to choose the grid point time series ($RC_l$) for which the variance is largest, and calculate its time derivative ($RC_l'$). The phase of the mode at each time step is determined by calculating the angle 30   between the vector ($RC_l$, $RC_l'$) and the vector $(0, 1)$. These phases, in the interval $(0, 2\pi)$, are then classified into eight equally populated categories. Composite maps are constructed from these categories.

## 3 Results

### 3.1 Reanalysis decompositions

The main outcome of the RMSSA method, the space-time principal components (ST-PCs) characterise both the spatial and temporal structure of the modes of variability. Sections 3.1 – 3.4 focus on their temporal aspects. The leading 30 ST-PC time series and the corresponding power spectra are displayed in Figure 2 for 20CR and ERA-20C, ordered according to the explained variance. We can see that

– components with predominantly multi-decadal periodicity (1, 2, 5, and 6) explain a total of 7.2 % and 5.9 % of the variance in 20CR and ERA-20C, respectively, with clear similarities in their time series and spectra

– multi-annual components (3, 4, 7, and 8) explain 4.2 % and 3.2 % of the variance in 20CR and ERA-20C, respectively

– there is a broad multi-annual peak centered at 5 yr and a narrower peak at 3.5 yr in both reanalyses; these are clearly separated in ERA-20C at the components 3 and 4 versus 7 and 8. This separation in 20CR is less clear

– there are many spectral peaks in the reanalyses at 2–3 year periods with little explained variance but some are well separated and distinct

The conclusion based on Figure 2 is that the leading sources of the near-surface air temperature variability at multi-decadal and multi-annual periods are well identifiable in the reanalysis data sets. 20CR and ERA-20C are composed of very similar components explaining the variance in the two data sets. This is of course expected but it is also reassuring from the methodological view point: despite its complexity, the RMSSA decomposition is consistent.

It is noteworthy in Figure 2 that the components 3, 4, 7, and 8 in both reanalyses have become more prominent with time. Since about the 1970's, the amplitudes of these 3.5 yr and 5 yr oscillations have been at a higher level, presumably due to some combination of forced climate change, intrinsic low-frequency climate variability, or changes in global observing network (the rather sudden increase in the amplitude seems to coincide with the onset of the modern era of satellite observations). This finding seems to be in support of e.g. Russell and Gnanadesikan (2014). In this connection it should be noted, however, that apparent low-frequency variations and changes in amplitude may simply arise from random fluctuations of the time series (Wunsch, 1999; Wittenberg, 2009). Back-projection of these components into the original grid representation (Figure 3), reveals that the components are indeed associated with the ENSO phenomenon and are geographically similar in 20CR and ERA-20C. In the snapshots from January 1987 and January 1998 (Figure 3), there is a typical El Niño pattern with positive anomalies in the equatorial Pacific Ocean, South-America, and northwestern North-America. These are associated with synchronous evolution of (i) a dipole structure in the western Antarctica with easterly motion, and (ii) a wave-train type pattern in the northernmost North-America with north-easterly motion. The components 3, 4, 7, and 8 thus represent a global phenomenon, with an increased amplitude in recent decades. These features are nicely depicted in our Youtube channel (https://www.youtube.com/watch?v=vehbT8fOHeM, https://www.youtube.com/watch?v=xG--SiUqqAI).

## 3.2 Reanalysis total spectra

Figure 4a shows the total spectrum for the reanalyses constructed from the ST-PCs, and their confidence intervals (dashed lines). As in the ST-PCs, there is most power in the slow modes. At periods of about 3.5 yr and 5 yr, there are the spectral peaks of the components 3, 4, 7, and 8. The dip at 1 yr reflects the removed annual cycle.

As Fig. 2 already suggests, the shape of the two spectra is remarkably similar in all time scales (Fig. 4a). This leaves hardly any doubt that the data assimilation systems of 20CR and ERA-20C extract observed information in a very similar manner. There are some differences, however. The spectral power in ERA-20C is systematically slightly higher than in 20CR. This difference is statistically significant at almost all time scales. This is most likely due to generally higher temperature variance in ERA-20C compared to 20CR, especially in the Southern Ocean and Arctic Ocean. Also, in 20CR, the 3.5 yr and 5 yr spectral peaks are relatively more pronounced than in ERA-20C.

Statistical significance tests are presented in Figs. 4b and 4c for 20CR and ERA-20C, respectively. The multi-annual periods (less than 7 yr) rising above the 95 % confidence interval (i.e., the red dots above the region covered by the vertical bars) are 3.5 yr, 3.6 yr, and 5.7 yr in 20CR and 3.6 yr, 5.2 yr, 5.5 yr, and 5.7 yr in ERA-20C. Thus, nearly the same periodicities rise above the red noise in the two data sets. It is logical that the frequency corresponding to the annual cycle is present in the red noise surrogates while it is absent from the data, and therefore the red dots fall far below their expected values. Interestingly, the period of 2.9 yr in 20CR and ERA-20C fall below the 95 % confidence interval. Our conclusion is therefore that the multi-annual climate variability in the near-surface air temperature is very similar in 20CR and ERA-20C.

## 3.3 CMIP5 model total spectra

The total spectra for the 12 CMIP5 model are shown in Fig. 5 (solid lines) with their 95 % confidence intervals (dashed envelopes) and the reanalysis spectra as a reference (thin lines). Statistically significant multi-annual modes (at 5 % level) are denoted by vertical dashed lines. As in the case of reanalyses, these spectra are unique expressions of the low-frequency variability present in the simulation data. A comparison between the simulated and the reanalysis spectra provides one means to assess the strengths and weaknesses of these models. However, one cannot simply rank the models based on how "far off" the model spectra are from the reference, because this comparison focuses on just one (although important) aspect of model performance and because seemingly good agreement with observations might occasionally result from compensating errors in model processes.

Here we will concentrate on the multi-annual aspects but note in passing that the level of multi-decadal variability (> 20 yr) is close to reanalyses in models a, c, d, e, and g. In the rest of the models, the level seems too low. In the decadal scale (∼10 ... 20 yr), the level of variance is close to reanalyses in a, b, c, f, i, j, and l. Subjectively, the shape of the low-frequency end of the spectra appears most realistic in models a and c.

In multi-annual scales, the model performance varies a lot among the models. There is a group of models (a, b, d, and e) with high spectral density at about 3 – 7 yr periods. The models d and e have a bi-modal spectral structure, as in the reanalyses, while

models a and b have a broad unimodal peak. Decompositions (available in the Supplementary material, S1) partly explain the reasons leading to these total spectra.

In model a, for instance, there is a unimodal broad peak at 3.5 – 4 yr periods (Fig. 5a). The decomposition reveals that there are, in fact, two well separated component pairs at 3.5 yr and 4 yr generating one merged peak to the total spectrum (Fig. S1a in the Supplement). A development hint is thus to investigate these modes which can help to better understand some underlying modelling deficiencies, and to keep monitoring how this aspect of model performance evolves in the future model upgrades. An additional concern in model a is the excessive spectral density at about 2 yr and 7 – 10 yr periods.

In model e, there is a bimodal total spectrum (Fig. 5e), although far too pronounced as compared with the reanalyses. The decomposition (Fig. S1e in the Supplement) reveals that the ST-PC components 1 – 10 (except 7–8) are all multi-annual and peak strongly and well in isolation at 3 yr, 3.5 yr, 4 yr, and 5 yrs, explaining together no less than 13.9 % of the total variance. The development hint for model e is thus to investigate the mechanisms behind the components 1 – 10 and thereby obtain guidance for improving the realism of simulations.

In most other models, the multi-annual variability is less prominent than in the reanalyses. In model c (Fig. 5c), on one hand, the decomposition points out (Fig. S1c in the Supplement) that there are about 12 ST-PC components with periods between 1.5 – 3 yrs leading to a total spectrum with a broad peak of 2 – 3 yr periods. These components tend to have very regular cycles, remotely resembling a coupled harmonic oscillator and seemingly missing the "offbeats" or true quasi-periodicity of the reanalyses. The task seems to be to find out reasons why model c produces too rapid and regular multi-annual variability. In model g (Fig. 5g), on the other hand, the leading ST-PC components 1 – 9 are on either decadal or multi-decadal periods and these overwhelm the total spectrum. It should be important to find out the causes for this accentuated variability, especially on the decadal scale.

Finally, Fig. 5 casts light on models' overall level of variability compared to reanalyses. Clearly, this level in model h (Fig. 5h) is low. Curiously enough, the leading ST-PC component pair in model h explains only 1.4 % of variance and peaks at 3.2 yr. This corresponds to the isolated peak in the total spectrum.

### 3.4 Significance of multi-annual modes in CMIP5 models

In the reanalyses (Fig. 4), only a few multi-annual periods rise above the red noise (three in 20CR and four in ERA-20C). They are at approximately 3.5 yr and 5 yr periods. For the CMIP5 models, the test results are available in the Supplementary material (S2). In Fig. 5, the multi-annual modes with periods less than 7 yrs at the 5 % significance level are denoted by dashed vertical lines.

In summary, there are 5 – 15 statistically significant periods in the models, except model k (Fig. 5k) with three and model g (Fig. 5g) with zero periods. The large number of significant periods (models d and e, for instance) can be explained, at least partly by the fact that the modes are quasi-periodic and the spectral density therefore appears on a range of frequencies. This manifests as excursion of the red-noise threshold on several adjacent frequencies. This is typical for models with large spectral power on certain time scales. In model l (Fig. 5l), for instance, there are two broad and distinct spectral peaks at about 3.5 yr and 6 yr periods, and many significant periods are gathered at these and nearby frequencies. In contrast, models f and h (and

to some extent model c) have several significant and distinct periods between 2 yr and 7 yr. In terms of number of significant modes, models a, i, j, and k seem to be closest to the reanalyses.

## 3.5 Spatial patterns of the 3-4 yr mode

ST-PC components can be represented in the original coordinate system as so called reconstructed components (RC) that can
be visualised. In this section, some visualisation results are presented and discussed.

In ERA-20C, there is a spectral peak at 3.5 yr period, which is significant at 5 % level (Fig. 4). This peak is due to the ST-PC components 7 and 8 with spectral density closely concentrated on this frequency (Fig. 2). Figure 6 depicts composite maps of each of the eight phases of the 3.5 yr mode in ERA-20C. Firstly, the mode is global with the largest temperature anomalies in the Pacific and North-America. Secondly, the mode contains tropical Pacific temperature anomalies, like in the
ENSO phenomenon (e.g. Kleeman, 2008). The cold (warm) maximum is in phase 1 (5) with the anomalies extending to the South-American continent. Thirdly, there are traveling temperature anomalies at high latitudes on both hemispheres. These are described next.

In phase 1 (Fig. 6), there is a small warm temperature anomaly in the North-Pacific (lon 160°W, lat 30°N). This pattern slowly moves northeast reaching Alaska in phase 5 and then gradually dissipating over the northernmost North-America in
phase 8 (and being visible still in phase 1). There is a very similar evolution of a cold anomaly starting in phase 5. At the same time, there is an oscillating temperature anomaly over the Eurasian continent in opposite phase. In Fig. 6, there is also a traveling temperature anomaly in the Southern Hemisphere. In phase 8 (Fig. 6), there is a cold anomaly over the Southern Ocean (lon 160°W). This strengthens, moves east, weakens, and crosses the Antarctic Peninsula in phase 4 and remains in the Weddell Sea until phase 7. Similarly, there is a warm anomaly in phase 4 (lon 160°W) with similar evolution as the cold one.

Next, 20CR and the CMIP5 model behaviour is studied. The 3.5 yr mode is significant in 20CR and ERA-20C. For the illustration, we have chosen component pairs from the model decompositions (Supplementary material Fig. S1) that have spectral peaks between 3 and 4 years and do not express substantial variability on other time scales. In most climate models, such a corresponding mode exists, except in models g and k. In model c this mode is not significant at 5 % level, but it is illustrated anyway. Supplementary material reveals how these modes are represented in different data sets (Fig. S3–S14). The
format is the same as in Fig. 6. A short summary is presented next.

In 20CR (Fig. S3), the anomalies are weaker compared to ERA-20C (Fig. S4). This is mainly because the 3 – 4 yr mode is distributed on two component pairs in 20CR whereas in ERA-20C it is concentrated on one pair. Nevertheless, similar although weaker signal is evident in 20CR, such as the northeast propagation of the North-Pacific temperature anomaly. (Note that in Fig. 3, the combination of components 3, 4, 7, and 8 produce highly similar global patterns for 20CR and ERA-20C.) A
prominent feature is also the opposite temperature anomalies in the northern Eurasia versus North-America. All models (Figs. S5–S14) produce a temperature anomaly to the equatorial Pacific Ocean (and South-America). The amplitude is larger and/or the area extends further to the west than in ERA-20C in six models (a, b, d, e, h, l). The anomaly pattern in the northwestern North-America is present in all the models to some extent. In the reanalyses, the anomaly is strictly confined to land areas but in most models, it is either somewhat misplaced or extends to the adjacent sea areas and the Eurasian continent. Models c, e,

and f produce the North-American pattern quite similar to reanalyses, and the northeast propagation is captured to some extent by models b, c, f, i, and l.

## 4   Discussion

We note that a substantial portion of variance at inter-annual to inter-decadal timescales can be attributed to "climate noise" associated with processes with timescales much shorter than the inter-annual scale (Wunsch 1999; Feldstein 2000). If the amplitude of the variability mode exceeds some noise threshold (such as red noise), then the variability mode is also likely driven by some process external to the atmosphere, in addition to the climate noise. For example, large part of the inter-annual atmospheric ENSO pattern is presumably driven by anomalies of tropical diabatic heating associated with sea surface temperature anomalies (Feldstein, 2000). We assume that for this reason the multi-annual patterns related to ENSO clearly exceed the noise threshold in the results of this study.

## 5   Conclusions

The aim of this study is to decompose the 20th century climate variability into its multi-annual modes, and to assess how these modes are represented by the contemporary climate models. To this end, two 20th century reanalysis data sets and 12 CMIP5 model simulations for years 1901–2005 of the monthly mean near-surface air temperature have been decomposed using Randomised Multi-Channel Singular Spectrum Analysis (RMSSA). The statistical significance of the identified modes has been estimated with Monte Carlo simulations. The main conclusions are as follows.

Spectral properties of the 20CR and ERA-20C reanalysis data appear remarkably similar. The most prominent forms of variability in both data sets are related to approximately 3.5 yr and 5 yr modes which are significant at 5 % level. The spectral power in ERA-20C is systematically slightly higher than in 20CR. The 3.5 yr mode is illustrated in more detail. In ERA-20C, the mode is associated with typical ENSO pattern of temperature anomalies in the equatorial Pacific Ocean, South-America, and northwestern North-America. On top of these, the mode also contains a northeast propagating temperature anomaly over the northernmost North-America, and another eastward propagating anomaly in the vicinity of western Antarctica. Since about the 1970's, the amplitude of this 3.5 yr global mode have increased.

None of the 12 coupled climate models closely reproduce all aspects of the reanalysis spectra, although some models represent many aspects well. For instance, the GFDL-ESM2M model has two nicely separated ENSO -related periods although they are relatively too prominent as compared to the reanalyses. Also, a number of models represent the propagating temperature anomalies at 3 – 4 yr time frame. Some suggestions are provided in the text for potential model development aspects.

There is an extensive Supplement available presenting the results in visual format for each reanalysis and model data set. In the future, relaxation of the uni-variate nature of the present study would seem a natural extension. This is now possible since the use of random projections allow efficient data structures preserving compression. Of special interest would be to study

behaviour of variables directly linked with atmosphere-ocean coupling processes, such as heat, momentum, and moisture fluxes over oceans.

## 6 Data and code availability

All data used in this study was downloaded from open sources. The RMSSA algorithm and the statistical significance testing
are implemented using GNU licensed free software from the R Project for Statistical Computing (www.r-project.org). Our implementation is available on request. The animations of the 3–4 yr mode are available for all data sets at
https://www.youtube.com/channel/UCu1zJdwJfLaXvfvTqsKCLHw.

## Appendix A: Randomised multi-channel singular spectrum analysis (RMSSA)

The RMSSA algorithm and the significance test is briefly presented here. The original data matrix is $\mathbf{X}_{N \times L}$, where the columns
are called *channels*. In case of gridded data set, $N$ represents the time steps and $L$ is the number of grid points. It is useful to think $N$ as the time steps when the *sample* of dimension $L$ is collected. The dimension reduction is a projection $\mathbf{X}_{N \times L} \to \mathbf{P}_{N \times k}$, where $L \gg k$. In other words, we preserve all samples but reduce the sample dimension from $L$ to $k$. The dimension reduction is performed in two steps: (1) generate a random matrix $\mathbf{R}_{L \times k}$, where the matrix elements are $r_{ij} \sim N(0,1)$ and column vectors of $\mathbf{R}$ are normalised to unit length, and (2) project $\mathbf{X}$ onto $\mathbf{R}$:

$$\mathbf{P}_{N \times k} = \mathbf{X}_{N \times L} \mathbf{R}_{L \times k}. \tag{A1}$$

The next step is to construct an augmented data matrix $\mathbf{A}$, which contains $M$ lagged copies of each channel in $\mathbf{P}$. In RMSSA, $M$ represents the lag window. $\mathbf{A}$ now has $Mk$ columns and $N' = N - M + 1$ rows. The singular value decomposition of $\mathbf{A}$ is

$$\mathbf{A} = \mathbf{U}\mathbf{D}^{1/2}\mathbf{V}^T \tag{A2}$$

The vectors of $\mathbf{U}$ are the eigenvectors of $\mathbf{Z} = \dfrac{1}{Mk}\mathbf{A}\mathbf{A}^T$ and $\mathbf{V}^T$ contains the eigenvectors of $\mathbf{C} = \dfrac{1}{N'}\mathbf{A}^T\mathbf{A}$. These vectors are orthogonal and often called space-time principal components (ST-PCs) and space-time empirical orthogonal functions (ST-EOFs), respectively. Note that the ST-EOFs are now in reduced space $k$. Diagonal elements of $\mathbf{D}$ are the eigenvalues of $\mathbf{C}$ or $\mathbf{Z}$. Finally, the eigenvectors (ST-EOFs) are calculated in the original $L$-dimensional space by

$$\mathbf{V} \approx \mathbf{A}_o^T \mathbf{U}(\mathbf{D}^{1/2})^{-1}, \tag{A3}$$

where $\mathbf{A}_o$ is the augmented matrix of the original data matrix $\mathbf{X}$. Note that the calculation of ST-EOFs in Eq. (A3) can be limited only to the eigenmodes of interest.

The ST-PCs can be represented in the original coordinate system by the reconstructed components, RCs (Plaut and Vautard, 1994; Ghil et al., 2002). This transformation is given by

$$rc_{le}(n) = \frac{1}{M_n} \sum_{m=I_n}^{J_n} u_e(n-m+1)v_{le}(m),$$  (A4)

where $u_e$ are the ST-PCs and $v_{le}$ are the ST-EOFs calculated in Eq. (A3) (the part of ST-EOF corresponding to channel $l$). $e$ is the index of the eigenmode that is calculated. The normalisation factor $M_n$ and the summation bounds $I_n$ and $J_n$ are given in Ghil et al. (2002) and for the central part of the time series ($M \leq n \leq N-M+1$) they are $(M, 1, M)$, respectively.

RMSSA with significance testing is briefly presented in the following. Testing the MSSA components against a red-noise null-hypothesis requires orthogonal input vectors, which are obtained by calculating first a conventional PCA and retaining a set of dominant PCs. Therefore some additional calculation steps are included in the RMSSA-algorithm:

SVD of lower dimensional matrix $\mathbf{P}$ is calculated to obtain the principal components (PCs, calculated as $\mathbf{UD}^{1/2}$). PCs fullfil the orthogonality constraint exactly. PCs, that explain large part of the variance of the data set (e.g. 50 first), are retained to obtain matrix $\mathbf{T}$, where the columns are the PCs. Next, the augmented matrix $\mathbf{A}_{PC}$ is constructed from $\mathbf{T}$ and SVD is calculated as in Eq. (A2).

Finally, a large number of red-noise processes (i.e. surrogate data sets) are generated, and the confidence limits for the MSSA eigenmodes are determined. This signicance test (Monte Carlo MSSA) is described in detail in Allen and Robertson (1996).

*Author contributions.* HJ suggested the study and mostly wrote the article. TS implemented the methods, performed all computations, and wrote data and method descriptions, JS supported the method development, and JR the climate model data analysis.

*Acknowledgements.* This research has been funded by the Academy of Finland (project number 140771), Academy of Finland Centre of Excellence Programme (project number 272041), and the Fortum Foundation (grant number 201500127).

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

**Table 1.** Climate models used in the study. For more details of the models, see Table 9.1. in Flato et al. (2013).

| Model ID | Model name | Modeling center | Country |
|:---:|:---:|:---:|:---:|
| a | CanESM2 | CCCMA | Canada |
| b | CESM1(CAM5) | NSF-DOE-NCAR | USA |
| c | CNRM-CM5-2 | CNRM-CERFACS | France |
| d | CSIRO-Mk3.6.0 | CSIRO-QCCCE | Australia |
| e | GFDL-ESM2M | NOAA GFDL | USA |
| f | GISS-E2-R | NASA GISS | USA |
| g | HadGEM2-ES | MOHC | UK |
| h | INM-CM4 | INM | Russia |
| i | IPSL-CM5B-LR | IPSL | France |
| j | MIROC-ESM | JAMSTEC/AORI/NIES | Japan |
| k | MPI-ESM-MR | MPI-M | Germany |
| l | MRI-CGCM3 | MRI/JMA | Japan |

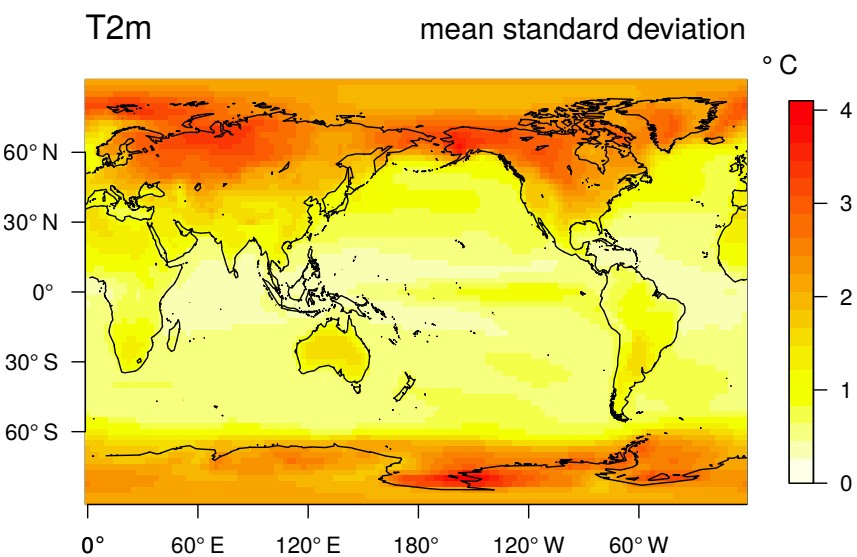

**Figure 1.** Map of the common normalisation factor. Shown is the mean standard deviation of 2 metre temperature (degC) across all the data sets.

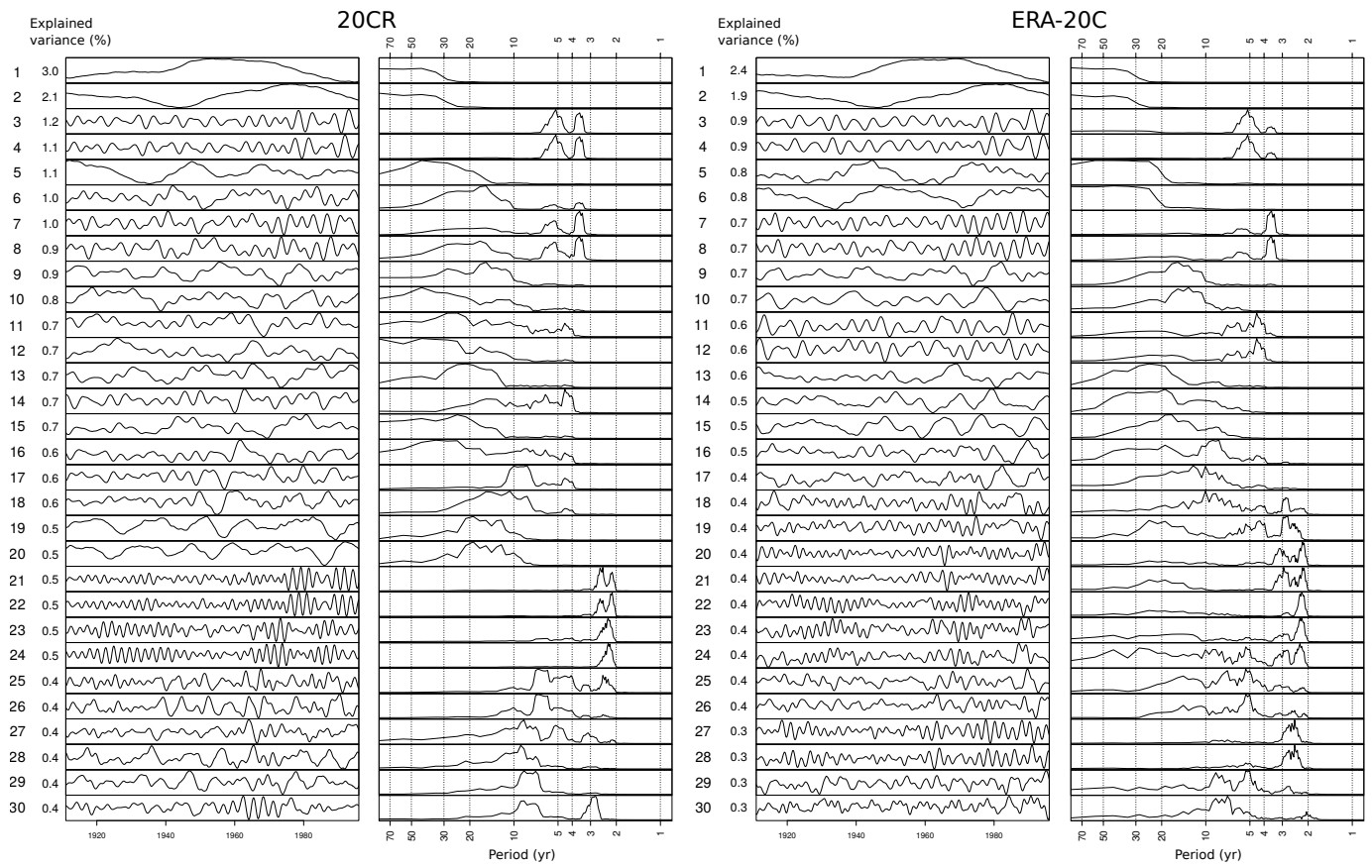

**Figure 2.** Reanalysis ST-PC time series (columns 1 and 3) of monthly near-surface temperature 1901–2005 and their spectra (columns 2 and 4) for 20CR and ERA-20C. The components are ordered according to the explained variance (%).

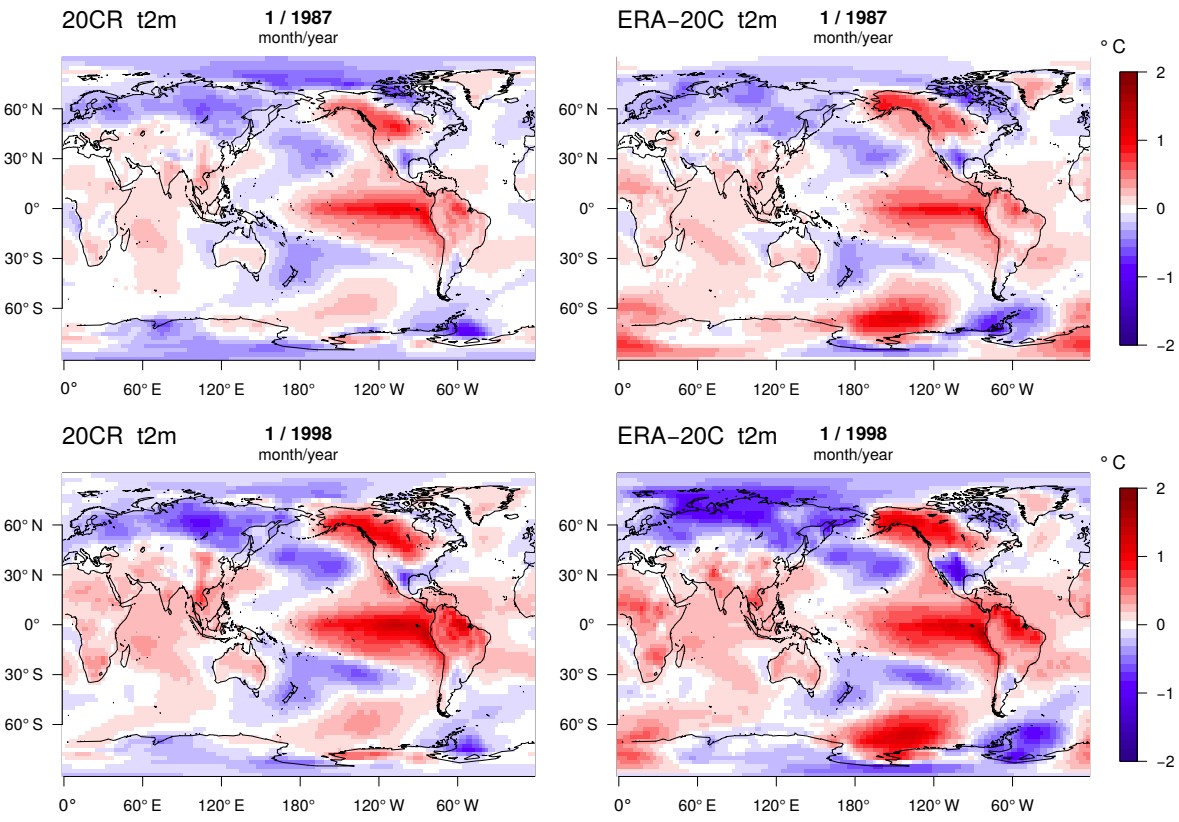

**Figure 3.** Global patterns of 2 metre temperature for the components 3, 4, 7 and 8 in 20CR (left column) and ERA-20C (right column). Snapshots are taken from Jan 1987 (top row) and Jan 1998 (bottom row). Unit degC.

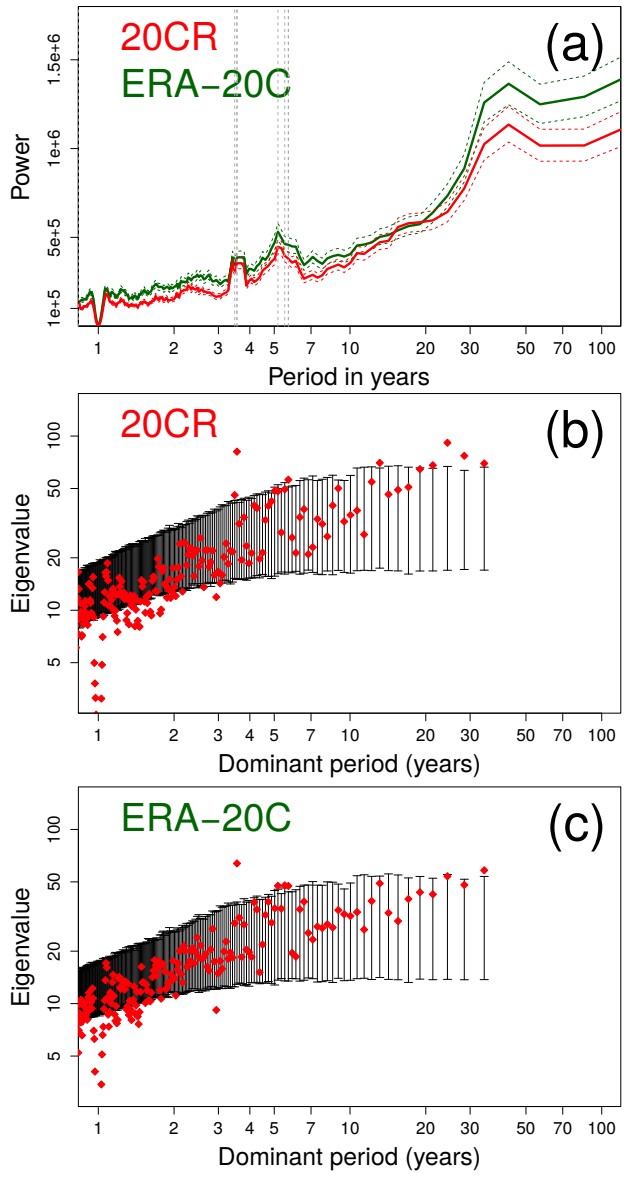

**Figure 4.** (a) Total spectrum of 20CR (red line) and ERA-20C (green line) with their min-max confidence intervals. The unit of the spectral density is arbitrary. (b) Significance of the 20CR periodicities against red-noise null-hypothesis. Shown are the data eigenvalues (red squares) and the $2.5^{th}$ and $97.5^{th}$ percentiles of the eigenvalue distribution of the red-noise surrogates (vertical bars). (c) Same as (b), but for the ERA-20C data set.

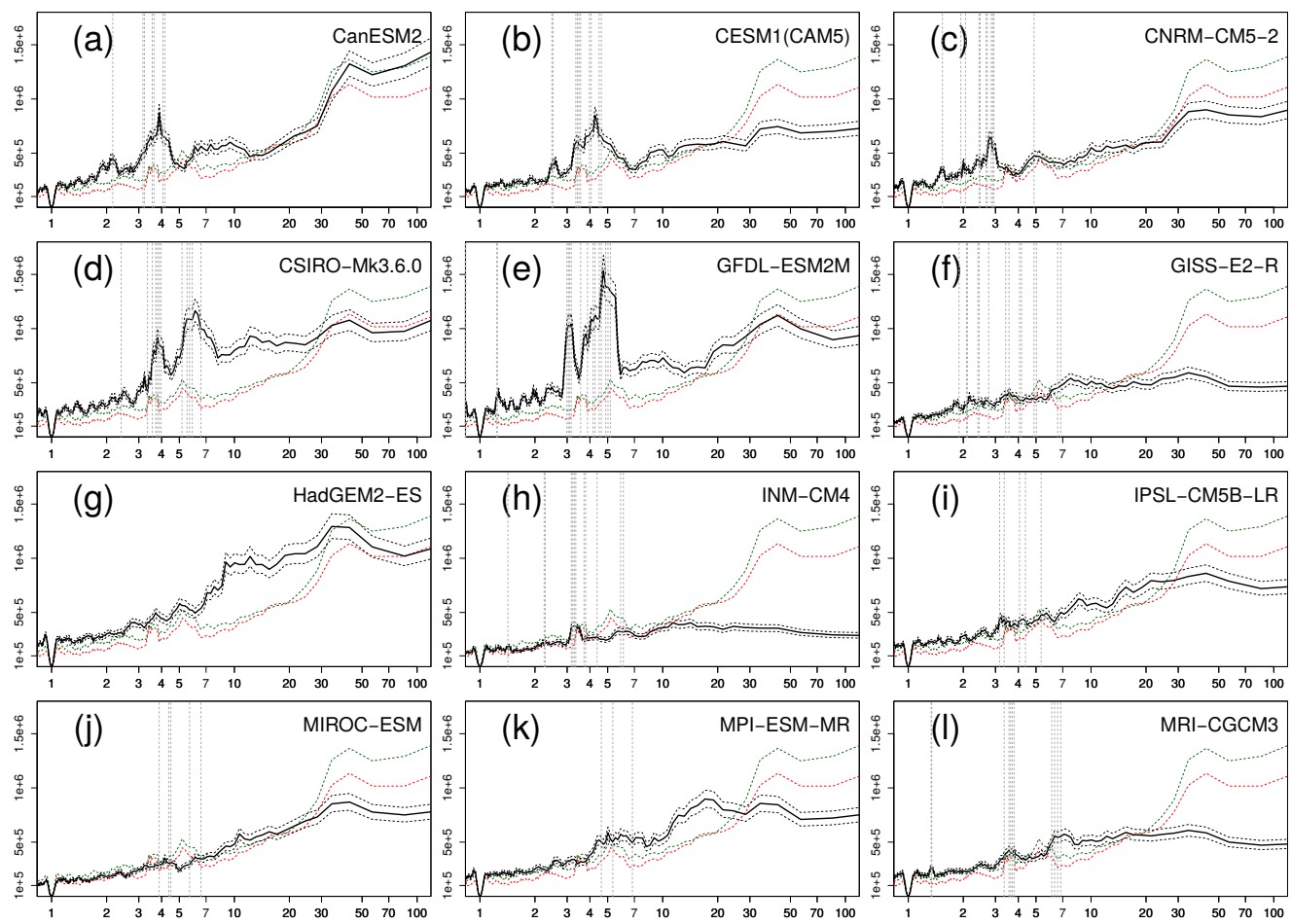

**Figure 5.** As Figure 4a but now for each climate model (black line). The reanalysis spectra are shown as a reference (dashed green and red lines). The dashed vertical lines indicate the climate model multi-annual periods significant at 5 % level.

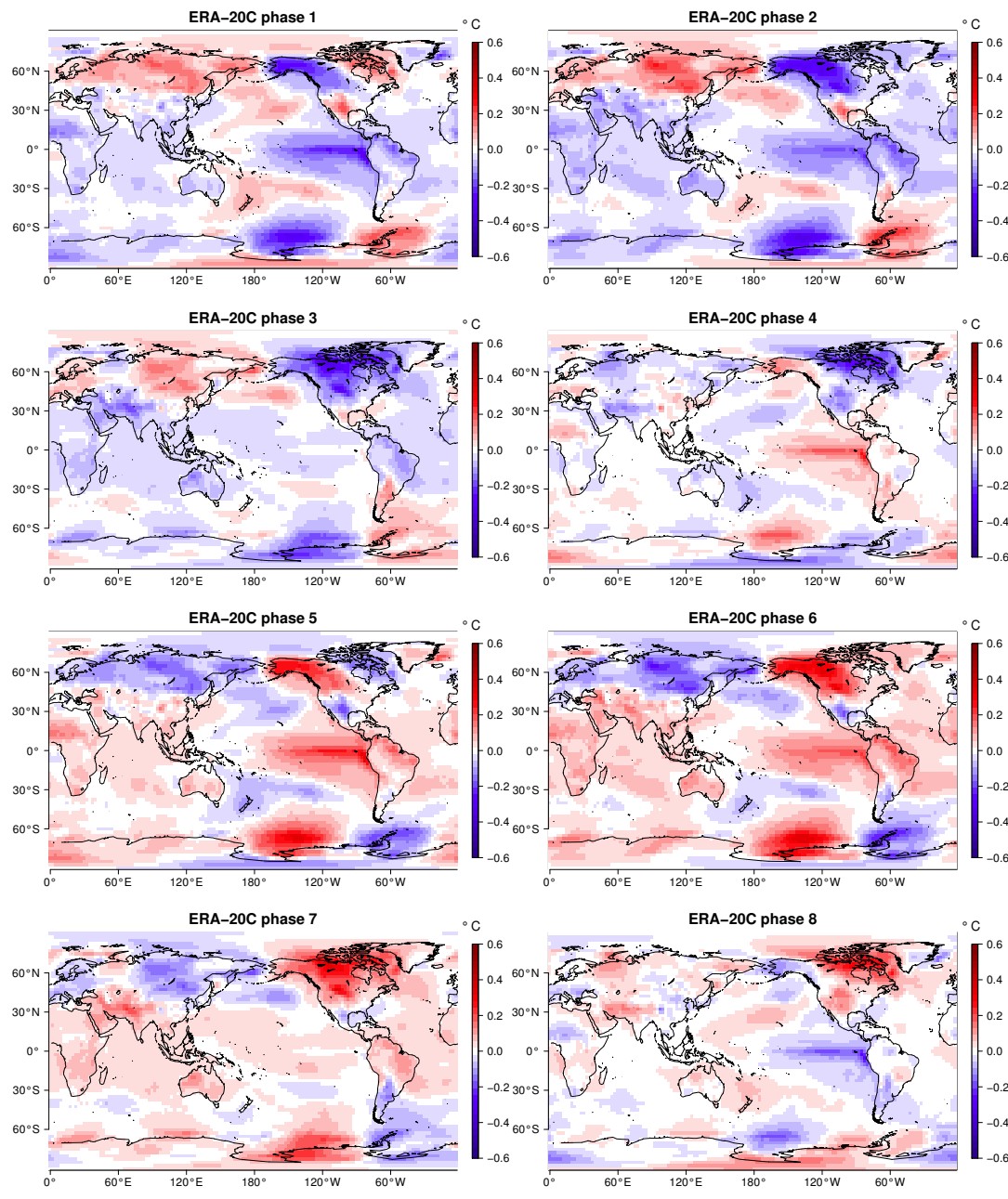

**Figure 6.** ERA-20C phase (1–8) composites of the 3–4 yr variability mode. Unit degC.