# Peer review of "Multi-annual modes in the 20th century temperature variability in reanalyses and CMIP5 models"

_Geoscientific Model Development, 2016_

## Referee Comment (RC1) · Anonymous Referee #1 · 29 Jun 2016

Review of "An evaluation of current capabilities of modelling low-frequency climate variability" by H. Järvinen et al.

This short study uses a recent statistical tool to explore interannual to decadal variability of near surface air temperature in 12 CMIP5 models.

Main comments: 1. The goal of this study unclear as it falls in between (1) a showcase of an advanced statistical tool (RMSSA) and (2) the evaluation of variability in CMIP5 models. Both goals have already been addressed at length in other publications and it is not clear what is new here. 2. The title seems to imply the second goal is pursued (model evaluation). Then it is unclear what the precise science question is. Why focus on these specific aspects of variability ? What implications for model use or development ? 3. The few lines that put in context model errors (p1/l19 to p2/l7) are quite weak

and provide an overly simplistic view of this complex problem. Also, why use only 12 models out of the 40+ CMIP5 model available ? 4. For ENSO time scale (and lower frequency), several studies have shown that a minimum of 200-300 years of simulation are necessary to obtain robust statistics (Wittenberg 2009 and Stevenson et al. 2010). This questions the use of historical simulations (140 years). 5. Spectra are not "objective" measures of model performance (nor any single metric, see IPCC AR5 Chap. 9) as error compensation can lead to the right statistics through the wrong balance of physical processes as shown in many studies. 6. The "subjective" discussions are quite vague, unhelpful and don't provide any perspective either compared to previous studies or for modelling groups. 7. Because of all these serious shortcomings, I recommend rejection of this manuscript.

References: Stevenson, S., Fox-Kemper, B., Jochum, M., Rajagopalan, B., & Yeager, S. G. (2010). ENSO Model Validation Using Wavelet Probability Analysis. Journal of Climate, 23(20), 5540–5547. doi:10.1175/2010JCLI3609.1 Wittenberg, A. T. (2009), Are historical records sufficient to constrain ENSO simulations?, Geophys. Res. Lett., 36, L12702, doi:10.1029/2009GL038710.

---

## Referee Comment (RC2) · Anonymous Referee #2 · 1 Aug 2016

This study aims to evaluate 12 CMIP5 models' performance in simulating climate variability using the RMSSA method. The authors examined the models' biases in reproducing the observed variance at different periods and in reproducing the observed spatial pattern of ENSO. The results appear to be interesting. The reviewer has several major concerns that need to be addressed.

1. Fig. 1 shows that the greatest variance is explained by decadal-multidecadal variabilities (after detrending). However, the decadal-multidecadal variabilities are not examined in this paper, including their spatial patterns and potential mechanisms as well as model biases.

2. Table 2: Some of the periods identified by the RMSSA are very close to each other (for example, 2.2, 2.3, and 2.5; 3.5 and 3.6). It is unclear whether those identified periods truly represent significantly different physical modes or they could merely represent the artifacts of the RMSSA method.

3. Figs. 1 and 3: In addition to ENSO, it will be useful to display the spatial patterns of other significant periods and examine the models' performance in simulating them.

4. Specific comments: a) Last paragraph of Page 1: Atmosphere's memory is too short to explain the signal with a period of 1.7 years. b) First paragraph of Page 2: Ocean dynamics responsible for the decadal-multidecadal variabilities needs to be discussed. c) Page 5, Line 19-20: Components 15-17 of ERA-20C appears to capture the decadal variability of ENSO. d) page 7, Line 16-25: Replace "a warm pool" and "a cold pool" by "a warm anomaly" and "a cold anomaly".

---

## Referee Comment (RC3) · Anonymous Referee #3 · 23 Aug 2016

This manuscript focuses on the capability of current climate models to simulate lowfrequency climate variability, as determined through a randomised multi-channel singular spectrum analysis (RMSSA) of near-surface air temperature. On the basis of this analysis, the authors conclude that state-of-the-art climate models tend to exhibit variability that is too periodic, under-active at multidecadal timescales, and over-active at decadal timescales. On the positive side, I thought that the manuscript was clearly and smoothly written. However, I was left with many questions about the authors' choices. The title of the manuscript is very broad and ambitious, but the authors only analyse one variable with one method in only 12 climate models, so any conclusions that are drawn are much narrower in scope than the title would suggest. By focusing on statistically significant periodicities, the authors really do not directly address whether or not models have too much or too little low-frequency variability (particularly since all time

series are standardized prior to the analysis). All comparisons between the models and reanalysis are informal and subjective, and all formal significance testing is limited to red noise null hypotheses rather than model/reanalysis differences. Overall, I was hoping that this study would provide a more thorough and objective evaluation of model performance that goes beyond previous studies, or if that was not the intention, that the scope of this study would be more clearly articulated. I describe my concerns more thoroughly below.

**Major Comments**

1) Lines 41-48: The attribution of variance at different timescales by the authors is too simple and not entirely accurate. A substantial portion of variance at interannual to interdecadal timescales can be attributed to "climate noise" associated with processes with intrinsic timescales that are much shorter than interannual. That is the nature of red noise. For example, the North Atlantic Oscillation (NAO) is a teleconnection pattern with broad impacts and pronounced interannual and interdecadal variability, and yet much of that can be attributed to internal atmospheric variability (Wunsch 1999; Feldstein 2000). Therefore, it is not accurate to say that interannual variability is primarily attributed to ENSO or that decadal-to-multi-decadal variability, but then the authors need to explain why they are focusing on oscillatory behavior and neglecting other dominant sources of interannual and multi-decadal variability.

Wunsch, C.: The interpretation of short climate records, with comments on the North Atlantic and Southern Oscillations. Bulletin of the American Meteorological Society, 80, 245-255, 1999.

Feldstein, S. B.: The timescale, power spectra, and climate noise properties of teleconnection patterns. Journal of Climate. 13, 4430-4440, 2000.

2) Why do the authors choose the 12 models that they choose? Given that there are so many more simulations available, this choice seems arbitrary.

3) Line 166: Again, perhaps this relates to my misconception about what the authors are trying to address, but the decision to standardize the data sets has made it challenging for me to interpret the authors' results. The climate models may have very different temperature standard deviations, which would impact the temperature variability from interannual to multidecadal timescales (e.g., Thompson et al. 2015). However, by standardizing the data, the authors essentially are artificially adjusting the climate models and reanalyses to have common variance at every grid point. Therefore, the authors are erasing potentially important differences in variance between the models and reanalyses that would impact reanalysis/model differences at all timescales. The motivation for this decision and the consequences for interpretation should be discussed.

Thompson, D. W. J., Barnes, E. A., Deser, C., Foust, W. E., and Phillips, A. S.: Quantifying the role of internal climate variability in future climate trends. Journal of Climate, 28, 6443-6456, 2015.

4) Lines 230-234: The decision to evaluate model performance subjectively is unsatisfying. It is difficult to compare power spectra with short records, and visual inspection can be deceiving. Combined with my previous comment, I have difficulty interpreting the authors' results. There may be truth in the authors' conclusions in lines 252-254, but I would like more support.

5) Line 272: How are "false alarms" defined? Again, the authors did not determine if there are significant differences between the reanalyses and models, and so I do not see how the determination of false alarms was made.

6) Line 289: Why did the authors subjectively choose the Nino3.4 region to base the composites? Although it seems reasonable that the 3.5-yr mode would be related to ENSO, by basing the composites on a subjectively chosen region, the authors seem to be predisposing the analysis to highlight ENSO-like variability.

More generally, I am not sure why Section 3.4 is entirely focused on ENSO and its

СЗ

teleconnections, given that these topics have been covered extensively in other studies and that the authors argue that five periodicities exist. It would seem less arbitrary to let the analysis direct the content and to focus on all identified periodicities.

Minor Comments

1) Lines 137-138: This relates to my first comment, but the authors are not really addressing whether the models "capture the observed temperature distribution."

2) Line 179: Is there any sensitivity to this choice of lag window?

3) Line 202: I do not understand why those components are called "trend components" if the data were detrended.

4) Lines 208-209: How is it determined that ENSO variability has a decadal component in 20CR?

5) Line 210: I would not consider the similarity of the 20CR and ERA-20C spectra to be striking, given that the reanalyses assimilate similar data.

6) Line 245: I am not convinced of five key periodicities. Physically, it seems that all identified periods may relate to one phenomenon (ENSO), and these five frequencies just happened to pass the significance threshold.

7) Discussion: Isn't it possible that the existence of too many significant periodicities in the climate models could be due to ENSO being too periodic in some models, which has been discussed previously?

---

## Author Response (AR1)

**Response to Referee #1**

We want to thank the three anonymous referees for the very thorough review of our manuscript. In particular, the comments helped us to better articulate the science question of the manuscript, and this hopefully resolves some of the major concerns. We shifted the focus of the paper from general low-frequency variability to multi-annual oscillations, and changed the title to "Multi-annual modes in the 20th century temperature variability in reanalyses and CMIP5 models".

The comments led to substantial changes in the manuscript. One of the main changes is that we have made is the way the data sets are preprocessed. We have now used a common scaling factor for all the data sets in order to be able to compare the total spectra of the data sets (based on the reasoning of Referee #3). Because of this comment, we have recalculated all results and also made substantial changes to the text, especially in the section describing the Results. Re-calculation did not change the big picture, but the results are now much better justifiable, especially as there is now a new Supplement available.

Because of these substantial changes, we kindly ask the Referees to read the whole manuscript once again.

We hope that these and the changes explained below help to better convey our message. Below are our detailed responses to the reviewer #1 (In the following, our response to each comment is in red font, and the referee's comment in black).

**(1) comments from referees/public**

1. The goal of this study is unclear as it falls in between (1) a showcase of an advanced statistical tool (RMSSA) and (2) the evaluation of variability in CMIP5 models. Both goals have already been addressed at length in other publications and it is not clear what is new here.

**(2) author's response**

Thank you for your remark. We totally agree that the goal was not clearly articulated in the original submission. We hope that the revised manuscript is more of (2) and less of (1). We hope the novel aspects are better conveyed in the revised manuscript so that it no longer unclear what is new here. Although there are a large number of studies on the evaluation of CMIP5 models, we still think that it is worthwhile to have a closer look at the model spectra, especially as the advanced tool (RMSSA) has not been applied never before in this extent in other publications.

**(3) author's changes in manuscript**

We have modified the title as well as the introduction of the manuscript to clarify the goal (which is to decompose the 20th century climate variability into its multi-annual modes, and to assess how these modes are represented by the contemporary climate models.)

**(1) comments from referees/public**

2. The title seems to imply the second goal is pursued (model evaluation). Then it is unclear what the precise science question is. Why focus on these specific aspects of variability ? What implications for model use or development?

**(2) author's response**

The title is changed, and from the revised manuscript it should be now very clear that we produce a reference decomposition from two reanalyses on multi-annual scales and then assess how the model data performs with respect to the reference. The science question is clarified, and we provide hints for model development, but refrain from speculating what exactly may be behind some model deficiencies.

Due to this, and comments by other Referees, we shifted the focus to multi-annual variability because of better statistical confidence of the results. We hope this to provide guidance for model development due

better understanding of the deficiencies in representing reanalysed modes of multi-annual climate variability.

**(3) author's changes in manuscript**

The comment induced a major revision of the text, especially in the Sections of Introduction and Results.

**(1) comments from referees/public**

3. The few lines that put in context model errors (p1/l19 to p2/l7) are quite weak and provide an overly simplistic view of this complex problem. Also, why use only 12 models out of the 40+ CMIP5 model available?

**(2) author's response**

We agree that the text was too simplistic, even though our goal was not to provide a comprehensive review of the complex question.

A subset of CMIP5 models was chosen to keep the analysis and presentation of results manageable. In selecting the models, a major principle was to use only one model per institution, so to avoid models that are too close relatives. Furthermore, all these models have undergone a long (generally several generations of) history of development, suggesting that the chosen models collectively represent the state-of-the-art.

**(3) author's changes in manuscript**

The explicitly mentioned lines are removed. A justification to the choice of the models have been added. We want to point out again that it would be advisable to read the manuscript once again, since the revisions have been quite extensive - we cannot simply point to a changed word here and a sentence there.

**(1) comments from referees/public**

4. For ENSO time scale (and lower frequency), several studies have shown that a minimum of 200-300 years of simulation are necessary to obtain robust statistics (Wittenberg 2009 and Stevenson et al. 2010). This questions the use of historical simulations (140 years).

**(2) author's response**

We agree that it would be ideal to have time series of 200 - 300 years to obtain robust statistics. In model simulation studies this is of course a possibility. However, the fact of reality is that the longest observation based references only extend over the past century, and this is what there is.

MSSA (and therefore also RMSSA) is especially designed for analysing short time series (see Ghil et al. 2002). By taking lagged copies of the time series, it provides overlapping views of the series and enhances the identification of signals from the noise. We have also estimated the likelihood of the identified patterns being generated only by red noise. This is done by the Monte-Carlo significance test, as described in the paper. The test shows that the multi-annual oscillations, have at most 5% chance of being generated only by red noise in both reanalysis datasets (Figure 4b and c) and most of the climate model simulation datasets (Figure 5). Therefore we can even argue that long time-series are in part needed because weak methods are used to analyse high-dimensional data.

**(3) author's changes in manuscript**

The focus is shifted to multi-annual scales and abstained from closer scrutiny of the decadal and multidecadal scales (please see the the text in p.2, I. 4-14.) Proposed references are added.

**(1) comments from referees/public**

5. Spectra are not "objective" measures of model performance (nor any single metric, see IPCC AR5 Chap.9) as error compensation can lead to the right statistics through the wrong balance of physical processes as shown in many studies.

**(2) author's response**

We agree that 'objective' was not the best word to use in this context. We agree that the spatio-temporal modes and their spectra are not objective performance metrics that allow ranking the models based on how different the model spectra are from the reference (reanalyses). However, we see that the total spectra and decompositions of each model provide useful hints of the strengths and weaknesses of the models.

We would like to point out, however, the differences of the method used here and the traditional spectrum analysis. RMSSA separates the variability modes that are independent of each other as orthogonal components, i.e. ST-PCs. We are then using spectrum analysis as a means to show on which frequency each component has most power. These spectra are then summarized to make the comparisons of the variability patterns in different data sets easier. We do not have to calculate any spatial averages to obtain the total spectra and also the regional differences in the variability patterns are included in the spatio-temporal analysis.

**(3) author's changes in manuscript**

The comment has been included and the text is now changed (Section 3.3, 1st para). Total spectra and decompositions (Supplement) of each model are now available and commented in the Results.

**(1) comments from referees/public**

6. The "subjective" discussions are quite vague, unhelpful and don't provide any perspective either compared to previous studies or for modelling groups.

**(2) author's response**

We agree, and this element has been removed altogether, and instead the revised manuscript now provides some perspective on the strengths and weaknesses of the models in simulating the multi-annual modes of temperature variability.

**(3) author's changes in manuscript**

Text changed as suggested in Results section.

**Response to Referee #2**

We want to thank the three anonymous referees for the very thorough review of our manuscript. In particular, the comments helped us to better articulate the science question of the manuscript, and this hopefully resolves some of the major concerns. We shifted the focus of the paper from general low-frequency variability to multi-annual oscillations, and changed the title to "Multi-annual modes in the 20th century temperature variability in reanalyses and CMIP5 models".

The comments led to substantial changes in the manuscript. One of the main changes is that we have made is the way the data sets are preprocessed. We have now used a common scaling factor for all the data sets in order to be able to compare the total spectra of the data sets (based on the reasoning of Referee #3). Because of this comment, we have recalculated all results and also made substantial changes to the text, especially in the section describing the Results. Re-calculation did not change the big picture, but the results are now much better justifiable, especially as there is now a new Supplement available.

Because of these substantial changes, we kindly ask the Referees to read the whole manuscript once again.

We hope that these and the changes explained below help to better convey our message. Below are our detailed responses to the reviewer #2 (In the following, our response to each comment is in red font, and the referee's comment in black).

**(1) comments from referees/public**

1. Fig. 1 shows that the greatest variance is explained by decadal-multidecadal variabilities (after detrending). However, the decadal-multidecadal variabilities are not examined in this paper, including their spatial patterns and potential mechanisms as well as model biases.

**(2) author's response**

The comment is exactly right: we did not provide many details about these slower modes although it would be very interesting to see some more details. The revised manuscript is even scarcer in this respect since due to the review comments, the scope is now firmly on multi-annual modes. We think the Referees' comments were justified (that there is no statistical significance in the results related to the slow modes) and followed the advice in scoping the manuscript anew. We only note briefly in the revised manuscript that the models behave quite differently regarding the variability in decadal and multi-decadal scales

**(3) author's changes in manuscript**

These are changes throughout the revised manuscript due to the refined scope, especially in Section 3.3, 2nd para.

**(1) comments from referees/public**

2. Table 2: Some of the periods identified by the RMSSA are very close to each other (for example, 2.2, 2.3, and 2.5; 3.5 and 3.6). It is unclear whether those identified periods truly represent significantly different physical modes or they could merely represent the artifacts of the RMSSA method.

**(2) author's response**

Thank you for this remark which is now addressed in the revised text. The identified modes in the reanalysis data and CMIP5 models are quasi-periodic, meaning that the oscillation is wobbly and within some neighborhood of a given frequency. Thus, more than one frequency in this neighborhood will be identified as significant. This seems to explain Table 2 of the original submission. In addition, the method itself has a certain spectral resolution depending on the analysis window and temporal resolution of the original data set (monthly data in this case).

Based on the review comments, we realised that Table 2 is not very reader-friendly, and is now removed. The information is now incorporated in Figure 5 instead, which is more compact regarding the significant multi-annual periods. Figure S2 in the Supplementary material provides the test results exhaustively.

**(3) author's changes in manuscript**

Text has been changed (Section 3.4), Table 2 is removed and the information is incorporated in Figure 5. Supplementary material added.

**(1) comments from referees/public**

3. Figs. 1 and 3: In addition to ENSO, it will be useful to display the spatial patterns of other significant periods and examine the models' performance in simulating them.

**(2) author's response**

We totally agree with this comment. The snag with this option is that there would soon be an excessive number of figures. In the revised manuscript, we selected to visualize a mode that is simulated reasonably well by most models, and the 3-4 yr variability pattern was the best option for this purpose.

**(3) author's changes in manuscript**

New figures available in the Supplement.

4. Specific comments:

**(1) comments from referees/public**

a) Last paragraph of Page 1: Atmosphere's memory is too short to explain the signal with a period of 1.7 years.

**(2) author's response**

This is true. In the revised manuscript, the explanatory power of the 1.7 yr mode has become weaker, presumably because of the new normalization with the common variance, and the mode no longer pop up so dramatically. There is thus no longer discussion about this mode in the revised manuscript.

**(3) author's changes in manuscript**

Text removed about the 1.7 yr mode.

**(1) comments from referees/public**

b) First paragraph of Page 2: Ocean dynamics responsible for the decadal-multidecadal variabilities needs to be discussed.

**(2) author's response**

This is completely true. However, with the focus on multi-annual modes, the revised manuscript no longer have this issue.

**(3) author's changes in manuscript**

No action taken.

**(1) comments from referees/public**

c) Page 5, Line 19-20: Components 15-17 of ERA-20C appears to capture the decadal variability of ENSO.

**(2) author's response**

The visualisation of all modes would be very nice. The new normalization with the common variance

changed the results somewhat especially regarding the modes with low explanatory power. Therefore the components 15-17 of ERA-20C were affected.

**(3) author's changes in manuscript**

No action taken.

**(1) comments from referees/public**

d) page 7, Line 16-25: Replace "a warm pool" and "a cold pool" by "a warm anomaly" and "a cold anomaly"

**(2) author's response**

Thanks, this is now corrected.

**(3) author's changes in manuscript**

Text is corrected.

**Response to Referee #3**

We want to thank the three anonymous referees for the very thorough review of our manuscript. In particular, the comments helped us to better articulate the science question of the manuscript, and this hopefully resolves some of the major concerns. We shifted the focus of the paper from general low-frequency variability to multi-annual oscillations, and changed the title to "Multi-annual modes in the 20th century temperature variability in reanalyses and CMIP5 models".

The comments led to substantial changes in the manuscript. One of the main changes is that we have made is the way the data sets are preprocessed. We have now used a common scaling factor for all the data sets in order to be able to compare the total spectra of the data sets (based on the reasoning of Referee #3). Because of this comment, we have recalculated all results and also made substantial changes to the text, especially in the section describing the Results. Re-calculation did not change the big picture, but the results are now much better justifiable, especially as there is now a new Supplement available.

Because of these substantial changes, we kindly ask the Referees to read the whole manuscript once again.

We hope that these and the changes explained below help to better convey our message. Below are our detailed responses to the reviewer #3 (In the following, our response to each comment is in red font, and the referee's comment in black).

**(1) comments from referees/public**

This manuscript focuses on the capability of current climate models to simulate low-frequency climate variability, as determined through a randomised multi-channel singular spectrum analysis (RMSSA) of near-surface air temperature. On the basis of this analysis, the authors conclude that state-of-the-art climate models tend to exhibit variability that is too periodic, under-active at multidecadal timescales, and over-active at decadal timescales. On the positive side, I thought that the manuscript was clearly and smoothly written. However, I was left with many questions about the authors' choices. The title of the manuscript is very broad and ambitious, but the authors only analyse one variable with one method in only 12 climate models, so any conclusions that are drawn are much narrower in scope than the title would suggest. By focusing on statistically significant periodicities, the authors really do not directly address whether or not models have too much or too little low-frequency variability (particularly since all time series are standardized prior to the analysis). All comparisons between the models and reanalysis are informal and subjective, and all formal significance testing is limited to red noise null hypotheses rather than model/reanalysis differences. Overall, I was hoping that this study would provide a more thorough and objective evaluation of model performance that goes beyond previous studies, or if that was not the intention, that the scope of this study would be more clearly articulated. I describe my concerns more thoroughly below.

**(2) author's response**

Thank you for this very thoughtful comment. It helped us, in fact, a lot to better formulate our thoughts and scope the revised manuscript better. It is clear that the manuscript title was too general compared to the actual content of our research, and there was a gap or discrepancy. We hope the revision has resolved this issue.

One of the main changes is the way the data sets are preprocessed. We have now used a common scaling factor for all the data sets in order to be able to better compare the total spectra of the data sets. We have re-calculated everything, including all figures, and made substantial changes to the text to accommodate this change.

**(1) comments from referees/public**

1) Lines 41-48: The attribution of variance at different timescales by the authors is too simple and not entirely accurate. A substantial portion of variance at interannual to interdecadal timescales can be attributed to "climate noise" associated with processes with intrinsic timescales that are much shorter than interannual. That is the nature of red noise. For example, the North Atlantic Oscillation (NAO) is a teleconnection pattern with broad impacts and pronounced interannual and interdecadal variability, and yet much of that can be attributed to internal atmospheric variability (Wunsch 1999; Feldstein 2000). Therefore, it is not accurate to say that interannual variability is primarily attributed to ENSO or that decadal-to-multi-decadal variability is attributed to ocean dynamics. These comments may be true for periodic variability, but then the authors need to explain why they are focusing on oscillatory behavior and neglecting other dominant sources of interannual and multi-decadal variability.

**(2) author's response**

Thanks for this clarification which we fully agree. We now realize that the original text was not entirely accurate. We have modified the introduction and these statements are not included anymore. Instead, we have added some text on this issue in the discussion on the lines of this comment (c.f. p. 10) and utilized the references.

**(3) author's changes in manuscript**

Introduction modified, and the statements removed. Text added on this issue in the discussion (c.f. p. 10), references are added.

**(1) comments from referees/public**

2) Why do the authors choose the 12 models that they choose? Given that there are so many more simulations available, this choice seems arbitrary.

**(2) author's response**

A subset of CMIP5 models was needed to keep the analysis and presentation manageable. In selecting the models, we used only one model per major institution to avoid models with close common ancestors. Furthermore, all these models have undergone a long history of development covering several model generations, suggesting that the chosen models collectively represent the state-of-the-art. We admit that the subset could be selected in many different ways.

**(3) author's changes in manuscript**

The choice of models is justified in the revised manuscript, and some text is added (Section 2.4, 1st para).

**(1) comments from referees/public**

3) Line 166: Again, perhaps this relates to my misconception about what the authors are trying to address, but the decision to standardize the data sets has made it challenging for me to interpret the authors' results. The climate models may have very different temperature standard deviations, which would impact the temperature variability from interannual to multidecadal timescales (e.g., Thompson et al. 2015). However, by standardizing the data, the authors essentially are artificially adjusting the climate models and reanalyses to have common variance at every grid point. Therefore, the authors are erasing potentially important differences in variance between the models and reanalyses that would impact reanalysis/model differences at all timescales. The motivation for this decision and the consequences for interpretation should be discussed.

**(2) author's response**

Thank you for this thoughtful remark. After quite some internal discussions we concluded that the way we normalized the data is not the best choice on the viewpoint of comparing the total spectra. We therefore

decided to recompute everything according to this comment about the standardisation, and use a common normalisation factor (the average standard deviation of all the data sets). This better retains comparability of total spectra. The revision of the manuscript is thus extensive (also including the new focus on multi-annual modes exclusively).

The data processing steps after the revision are:

- linear trend fitted and removed,
- annual cycle estimated using Seasonal-Trend Decomposition (STL; Cleveland et al., 1990) and removed,
- resulting values mean-centered and divided by the average standard deviation of all the data sets (see Figure 1). Average standard deviation is obtained after removal of the trend and the annual cycle.

These changes in the preprocessing has led to changes in the results, analysis and conclusions (not so much in the leading modes of variability but more of the modes of smaller eigenvalues). We note that the common normalisation factor may not be the optimal for each data set, but it supports better the aim of this study, which is to compare the multi-annual modes in reanalysis and climate model data sets.

**(3) author's changes in manuscript**

Text revised extensively.

**(1) comments from referees/public**

4) Lines 230-234: The decision to evaluate model performance subjectively is unsatisfying. It is difficult to compare power spectra with short records, and visual inspection can be deceiving. Combined with my previous comment, I have difficulty interpreting the authors' results. There may be truth in the authors' conclusions in lines 252-254, but I would like more support.

**(2) author's response**

We agree with this difficulty, and are not completely satisfied with the subjectivity either. The revised manuscript is our attempt for more objective conclusions. We have also included Supplementary material to better support the analysis. We have removed Table 3 and changed/removed the associated discussions. The conclusions are modified for objectivity, and solely focusing on the multi-annual variability.

**(3) author's changes in manuscript**

Table 3 is removed, and the associated discussions changed/removed. The conclusions are modified, and focus changed to multi-annual variability. A Supplement added.

**(1) comments from referees/public**

5) Line 272: How are "false alarms" defined? Again, the authors did not determine if there are significant differences between the reanalyses and models, and so I do not see how the determination of false alarms was made.

**(2) author's response**

Thanks for this comment - it helped us to realize that the original choice of making a subjective evaluation of model performance inevitably leads to this cascade of problems. We agree that 'false alarms' were not defined at all.

We do not use the term 'false alarm' in the revised manuscript. In addition, we have decided to remove Table 2 and show the significant multi-annual modes in Figure 5 (thin vertical lines) and also in the Supplementary material (S2). We think that these figures are more reader-friendly than Table 2, and the discussion more objective.

**(3) author's changes in manuscript**

The term 'false alarm' is removed, Table 2 is removed, and a Supplement is added.

**(1) comments from referees/public**

6) Line 289: Why did the authors subjectively choose the Nino3.4 region to base the composites? Although it seems reasonable that the 3.5-yr mode would be related to ENSO, by basing the composites on a subjectively chosen region, the authors seem to be predisposing the analysis to highlight ENSO-like variability. More generally, I am not sure why Section 3.4 is entirely focused on ENSO and its teleconnections, given that these topics have been covered extensively in other studies and that the authors argue that five periodicities exist. It would seem less arbitrary to let the analysis direct the content and to focus on all identified periodicities.

**(2) author's response**

It is true that the choice of the Nino3.4 region to base the composites directs the analysis (which is not nice), although the choice was made "post mortem", i.e., after inspecting all the individual figures, which seem to illustrate ENSO-type variability.

Inspired by your comment, we used a completely different and fully objective approach in the revised manuscript, which results in "phase composites". The compositing procedure now follows the one described in Plaut and Vautard (1994). The idea is to choose the grid point time series (RCi) for which the variance is the largest, and calculate its time derivative (RC'i). The phase of the mode at each time step is determined by calculating the angle between the vector (RCI, RC'i) and the vector (0, 1). These phases, in the interval (0,  $2\pi$ ), are then classified into eight categories, each occupied by equal number of "maps". Composite maps are then constructed from the maps in each category. This description is included in a new subsection (2.6 Data visualisation).

In the revised manuscript we have identified significant multi-annual periods in the reanalysis data sets at 3.5/3.6 and 5.2-5.7 yr. A variability mode with a period between 3 and 4 years was identified significant (at 5% level) in majority of the climate models (excluding models c, g and k) and therefore we decided to illustrate this particular mode.

There are indeed interesting but different patterns in several CMIP5 models that would be worth studying, but inclusion of all these would make this paper excessively long, and are therefore not included. We tend to think that the model development groups should do this for sake of their own model development.

**(3) author's changes in manuscript**

Results are recomputed and text changed extensively.

**Minor Comments**

**(1) comments from referees/public**

1) Lines 137-138: This relates to my first comment, but the authors are not really addressing whether the models "capture the observed temperature distribution."

**(2) author's response**

As far as we understand correctly this comment, the total spectra are now better inter-comparable, and therefore one can better assess the "capture" of variability.

**(3) author's changes in manuscript**

No action taken.

**(1) comments from referees/public**

2) Line 179: Is there any sensitivity to this choice of lag window?

**(2) author's response**

The sensitivity was studied in Seitola et al. (2015), c.f. Fig 7. In that paper it was concluded that the choice of the lag window does not have major effects on the significant periods (on multi-annual scales). We did not redo this sensitivity test here.

**(3) author's changes in manuscript**

No action taken.

**(1) comments from referees/public**

3) Line 202: I do not understand why those components are called "trend components" if the data were detrended.

**(2) author's response**

Sorry, calling the slow component as 'trend components' is a convention that has been used in some of our more statistics oriented references. We agree that this is misleading and the term has been replaced in the revised manuscript.

**(3) author's changes in manuscript**

Terminology is changed.

**(1) comments from referees/public**

4) Lines 208-209: How is it determined that ENSO variability has a decadal component in 20CR?

**(2) author's response**

In the original manuscript, Figure 1, the components 5 and 6 of 20CR have also spectral power between 10 and 20 yr periods, in addition to power on multi-annual time-scales. In the revised manuscript, Figure 2, a similar pattern is seen in components 7 and 8.

**(3) author's changes in manuscript**

No action taken.

**(1) comments from referees/public**

5) Line 210: I would not consider the similarity of the 20CR and ERA-20C spectra to be striking, given that the reanalyses assimilate similar data.

**(2) author's response**

We agree to some extent, but also think that this shows that the data assimilation systems of 20CR and ERA-20C extract observed information in a very similar manner (which is of course good news).

**(3) author's changes in manuscript**

The word "striking" does not appear in the revised manuscript.

**(1) comments from referees/public**

6) Line 245: I am not convinced of five key periodicities. Physically, it seems that all identified periods may relate to one phenomenon (ENSO), and these five frequencies just happened to pass the significance threshold.

**(2) author's response**

Thanks for the remark. We agree that the identified periods may relate to ENSO. Since it is a quasi-periodic oscillation, the ENSO-variability is captured by several near-by frequencies in the significance test.

**(3) author's changes in manuscript**

Text has been amended (Section 3.4).

**(1) comments from referees/public**

7) Discussion: Isn't it possible that the existence of too many significant periodicities in the climate models could be due to ENSO being too periodic in some models, which has been discussed previously?

**(2) author's response**

We agree that too strong / periodic ENSO may result in a large number of significant periodicities in climate models, and the significance test then to pick these up.

**(3) author's changes in manuscript**

The text has been amended in Section 3.4

**List of changes in the manuscript:**

We want to point out that it would be advisable to read the manuscript once again, since the revisions have been quite extensive - we cannot simply point to a changed word here and a sentence there. Here are the major revisions that we have made:

- We have changed the title to: "Multi-annual modes in the 20th century temperature variability in reanalyses and CMIP5 models"
- We have modified the Introduction to clarify the goal of the work (which is to decompose the 20th century climate variability into its multi-annual modes, and to assess how these modes are represented by the contemporary climate models.)
- The focus is shifted to multi-annual scales and abstained from closer scrutiny of the decadal and multi-decadal scales (please see the the text in p.2, I. 4-14.). There are changes throughout the revised manuscript due to the refined scope.
- One of the main changes is the way the data sets are preprocessed. We have now used a common scaling factor for all the data sets in order to be able to better compare the total spectra of the data sets. This is described in 2.5, p.4-5. We have re-calculated everything, including all figures, and made substantial changes to the text to accommodate this change.
- Change in compositing procedure: we used a completely different and fully objective approach in the revised manuscript, which results in "phase composites". The compositing procedure now follows the one described in Plaut and Vautard (1994). The idea is to choose the grid point time series (RCi) for which the variance is the largest, and calculate its time derivative (RC'i). The phase of the mode at each time step is determined by calculating the angle between the vector (RCI, RC'i) and the vector (0, 1). These phases, in the interval (0, 2π), are then classified into eight categories, each occupied by equal number of "maps". Composite maps are then constructed from the maps in each category. This description is included in a new subsection (2.6 Data visualisation). Please see the Figure 6 in the revised manuscript and also the supplement (S3).
- We have added a supplement including decompositions (S1), significance tests (S2) and spatial patterns of 3-4 yr mode for each data set (S3).
- A justification to the choice of the models have been added (section 2.4 data sources)
- Table 2 is removed and the information (significance test) is incorporated in Figure 5 and supplement (S2).
- Table 3 is removed, and the associated discussions changed/removed. The discussion (Section 4, p. 10) and conclusions (Section 5, p. 10) are modified, and focus changed to multi-annual variability.
- Figures 1, 3, and 6 are new
- Figures 2 (fig.1 in original version), 4 (fig. 2 in original version) and 5 (fig. 3 in original version) are modified

**An evaluation of current capabilities of modelling low-frequency climate Multi-annual modes in the 20th century temperature variability in reanalyses and CMIP5 models**

Heikki Järvinen1, Teija Seitola1,2, Johan Silén2, and Jouni Räisänen1

1Department of Physics, University of Helsinki, Finland

2Finnish Meteorological Institute, Helsinki, Finland

Correspondence to: Heikki Järvinen (heikki.j.jarvinen@helsinki.fi)

Abstract. A crucial performance test of performance expectation is that Earth system models is their ability to simulate simulate well the climate mean state and variability. Here we concentrate on representation of inter-annual to multi-decadal variability in 12 CMIP5 climate model simulations. Reference climate is provided by the climate variability. To test this expectation, we decompose two 20th century reanalysis data sets of and 12 CMIP5 model simulations for years 1901 –

- 5 2005 of the monthly mean near-surface air temperature . The spectral decomposition is based on using Randomised Multi-Channel Singular Spectrum Analysis (RMSSA). Due to the relatively short time span, we concentrate on the representation of multi-annual variability which the RMSSA method effectively captures as separate and mutually orthogonal spatio-temporal components. This decomposition is a unique way to separate statistically significant quasi-periodic oscillations from one another in high-dimensional data sets.
- 10 The main results are as follows. First, the total spectra for the two reanalysis data sets are remarkably similar in all time scales, except that spectral power of decadal variability (10–30 yr) differ in these data by about 30 the spectral power in ERA-20C is systematically slightly higher than in 20CR. Apart from the slow components related to multi-decadal periodicities, ENSO oscillations with approximately 3.5 yr and 5 yr periods are the most prominent forms of variability in both reanalyses. In 20CR, these are relatively slightly more pronounced than in ERA-20C. Since about the 1970's, the amplitudes of the 3.5 yr and 5 yr
- 15 oscillations have increased, presumably due to some combination of forced climate change, intrinsic low-frequency climate variability, or change in global observing network. Second, none of the 12 coupled climate models closely reproduce all aspects of the reference reanalysis spectra, although some models represent many aspects well. For instance, the IPSL-CM5B-LR model is close to reanalyses but has too little multi-decadal variability, and the HadGEM2-ES model is close to reanalyses except the notable over-activity at periods at and around 10 yrGFDL-ESM2M model has two nicely separated ENSO periods
- 20 although they are relatively too prominent as compared with the reanalyses. There is an extensive Supplement and Youtube videos to illustrate the multi-annual variability of the data sets.

*Keywords:* climate model assessment, dimensionality reductionspatio-temporal modes, climate variability, climate model simulation, random projection, 20th century reanalysis, significance testing, RMSSA algorithmRMSSA algorithm, ENSO oscillation, Youtube video

**25 1 Introduction**

The ultimate goal in developing Earth system models (ESM) is to exploit the inherent predictability of the Earth system to enable exploitation of the inherent Earth system predictability, and hence reduce weather and climate related uncertainties in our daily life, and guide societies in making sustainable choices (e.g., Slingo and Palmer 2011; Meehl et al. 2014). Prediction tools are very complex and their testing goes hand-in-hand with their development. A crucial performance test of ESMs is

30 related to their ability to simulate well the observed For the predictions to be useful and usable, the expectation is that the climate mean state and the variability around the mean.

Here we focus on ESMs of today and how they represent inter-annual to multi-decadal climate variability. This is a very broad range of temporal scales and it is associated with a multitude of spatial scales. Generally speaking, spectral misrepresentations appear either due to lack of variability in a model or over-activity of a model in some temporal scales.

- 35 Conclusions about model deficiencies based on spectral differences are very scale dependent, and some general guidance can be obtained by thinking about the mechanisms of natural climate variability (e.g., Ghil 2002). Essentially, short time scale variability (below 2 yr) in the model spectrum of near-surface air temperature is most likely related to the representation of internal variability of the atmosphere. Associated model deficiencies, such as low resolution, can explain most of these weaknesses. Inter-annual variability (2–7 yr) is prominently related to the ENSO phenomenon, and simulation weaknesses
- 40 point more towards deficiencies in atmosphere-ocean feedback processes and ocean model dynamics. Decadal-to-multi-decadal variability can be thought of being driven by ocean dynamics. There is however clear indication of multi-decadal variability, such as Atlantic multi-decadal oscillation (AMO), that may be driven by stochastic forcing of mid-latitude atmospheric circulation on ocean, and changes in ocean circulation may rather be a response rather than driver of the variability (Clement et al., 2015). This interpretation widens the scope of possible root causes from model errors in ocean dynamics to coupling
- 45 issuesclimate variability are well simulated by these tools. Due to the complexity of the models and the data they produce, testing the expectation poses a challenge: many aspects of the model performance are gathered under the variability concept and no single diagnostic alone is sufficient to exhaust its all facets. Yet, understanding the discrepancies between the observed and simulated variability is crucial feedback for model development.

Representation of inter-annual to multi-decadal climate variability among models participating in climate model inter-

- 50 comparisons(, such as CMIP5), has been studied by e.g. Bellenger et al. (2014), Knutson et al. (2013), Ba et al. (2014), and Fredriksen and Rypdal (2016). We will add to this literature by applying a recently developed powerful spectral analysis tool in this field. We identify the spectral signatures by interfacing a representative set of contemporary coupled climate models with reanalysis data focusing on spatio-temporal modes of climate variability. One century covered with global reanalysis data is naturally very short for this purpose and severely constrains inter-comparison studies (e. g. Wittenberg 2009 and Stevenson
- 55 et al. 2010). First, time series should cover a sufficient number of recurring "events" for obtaining significance for the findings. Therefore, decadal-to-multi-decadal variability is of interest but not as informative as focusing on shorter cycles of variability. Second, the applied methods have to be very effective in extracting information from the short but high-dimensional data sets. For these reasons, we concentrate on the representation of multi-annual variability in reanalyses and coupled climate models

applying Randomised Multi-Channel Singular Spectrum Analysis (RMSSA; Seitola et al. 2014, 2015) which is an advanced

- 60 effectively separates mutually orthogonal spatio-temporal components from our high-dimensional data sets.
- The aim of this study is to decompose the 20th century climate variability into its multi-annual modes, and to assess how these modes are represented by the contemporary climate models. We hope this to provide guidance for model development due better understanding of the deficiencies in representing reanalysed modes of multi-annual climate variability. Ultimately, interpreting the hints about model deficiencies as development topics are due for the development teams themselves. Our role
- 65 is to point towards the potential error sources. For reassuring the teams that high-dimensional time series analysis method for is possible today, we emphasise the methodological aspect of this study. RMSSA can, under very weak assumptions on the data, decompose high-dimensional problems. The strength of RMSSA lies in the fact it is able to data sets in a unique way and separate statistically significant quasi-periodic spatio-temporal oscillations from one another. This is in contrast to many other approaches which either make assumptions about the oscillation structures, such as Fourier or spherical decomposition,
- 70 or resolve only either spatial or temporal aspects of variability. RMSSA can detect spatially evolving "chains of eventsin high-dimensional systems by "through resolving eigenmodes of spatio-temporal covariance data. This is a significant advantage, say, over PCA which only resolves eigenmodes of spatial covariances . This can lead to undesirable projection of and often projects temporal evolution of an event "event" onto a number of different eigenmodes. Additional benefits of RMSSA are: (i) dimension reduction via In addition, the novel data compression based on random projections enable applications in
- 75 extremely high dimensional problems, (ii) convergence properties of the eigenmode decomposition are very good allowing better physical interpretation of fewer components, and (iii) the resulting spectrum is straightforward to test for significance. The paper is organised as follows: data and methods are explained in Section 2, results in Section 3, followed by discussion and conclusionshere 
[revised manuscript text omitted]
) 20CR has ~2.0 degree horizontal resolution and we have used gaussian gridded (192 × 94) data from 3-hour forecast values. The horizontal resolution of ERA-20C is approximately 125 km (T159) in a grid of 360 × 181 points and the reanalysis data sets were interpolated into the same grid as the model simulations (144 × 73 points).

**150 2.5 Data processing**

Some pre-processing of the data sets is was needed before applying RMSSAand statistical significance testing. The data sets were standardised (i.e. the time series of . At each grid point was mean-centered and divided by its standard deviation) to avoid overweighting the grid points with higher variance. This adds weight on the lower latitude variability, where ENSO-type variability is pronounced. On the other hand, no-scaling would make the higher latitude variability dominant because of larger amplitude variations there. Furthermore, each data set was de-trended, and the dominating the data sets were processed as

**follows:**

155

- linear trend was fitted and removed,

- annual cycle was estimated using Seasonal-Trend Decomposition (STL; Cleveland et al., 1990) and removed from the original time series.
- 160

180

resulting values were mean-centered and divided by the average standard deviation of all the data sets (see Figure ??).
 Average standard deviation was obtained after removal of the trend and the annual cycle.

The reanalysis and climate model data sets have different temperature standard deviations, which would impact the temperature variability from inter-annual to multi-decadal timescales (e.g., Thompson et al. 2015). To retain these differences, we have used a common normalization factor (i.e., the average standard deviation of all the data sets). This procedure reduces the weight

- 165 of grid points with high variance, typically at higher latitudes, and hence adds weight on the lower latitude features. After the pre-processing, the dimension reduction step of RMSSA was applied so that approximately 3-5% of the original dimensions of 20CR data, 0.8 of ERA-20C, and about 5 of climate model data were retained the different percentages are due to different volumes of the original datadata dimensions were retained. The lag window in the analysis was 20 yr (240 months). The total spectra was were obtained from this analysis, and are comparable due to normalisation using the common standard deviation
- 170 of the data sets.

In the The statistical significance test , the uses a red noise null hypothesis. In the test we have used data sets that are normalised by their own standard deviations. Using a common normalisation interferes with generating the red noise surrogates corresponding to each data set. The first 50 PCs of each data set were retained as input. Those PCs explain 80-79 % of the variability in 20CR, 75 % in ERA-20C, and 70 %–80 % in the climate model data sets. A total of 1000 realisations of red noise

175 surrogate data sets were generated, and confidence intervals (90 and interval (95 %) for the oscillatory modes were estimated. We note that transformation to PCs may intefere interfere with the detection of weak signals, as demonstrated by Groth and Ghil (2015).

In the following section we will compare the spectral properties of the reanalysis and model data sets. Furthermore, we will test the spectra of each data set against a red noise null hypothesis in order to distinguish signal from noise. Finally, we will compare the spatial patterns of an oscillatory modewith a

**2.6 Data visualization**

We used reconstructed components (RC; see Appendix ??) for visualisation of the spatial patterns related to ST-PCs. For each grid point time series, we can calculate the RCs corresponding to the ST-PCs (or modes) of interest. These RC values, reflecting the contribution of each grid point to the mode, can be plotted on a map at each time step. We have used these maps to

185 construct videos of the spatio-temporal modes. In Section 3.5yr period as represented by different data sets., we have analysed RCs corresponding to 3–4 yr variability. The result is a time series of the data corresponding to the 3–4 yr mode in each grid point and according to its variance after detrending and removing the annual cycle. In the analysis we have neglected 5 yrs in the beginning and the end of the time series, because the reconstruction procedure may be biased there (see the Appendix, eq. A4). The videos can be found at our Youtube channel (https://www.youtube.com/channel/UCu1zJdwJfLaXvfvTqsKCLHw).

To summarise the animations, we have calculated composite maps of the modes. The compositing procedure follows the one described in Plaut and Vautard (1994). The idea is to choose the grid point time series  $(RC_l)$  for which the variance is largest, and calculate its time derivative  $(RC'_l)$ . The phase of the mode at each time step is determined by calculating the angle between the vector  $(RC_l, RC'_l)$  and the vector (0, 1). These phases, in the interval  $(0, 2\pi)$ , are then classified into eight equally populated categories. Composite maps are constructed from these categories.

**195 3 Results**

**3.1 Spectral similarity of the two reanalysis data sets**

We first demonstrate the spectral similarity of the two reanalysis data sets. Figure **??** displays, in terms of explained variance, the leading 30-

**3.1 Reanalysis decompositions**

- 200 The main outcome of the RMSSA method, the space-time principal components (ST-PCsfor-) characterise both the spatial and temporal structure of the modes of variability. Sections 3.1 3.4 focus on their temporal aspects. The leading 30 ST-PC time series and the corresponding power spectra are displayed in Figure ?? for 20CR and ERA-20Cand the corresponding power spectra. The decomposition reveals that variance is distributed in a very similar way in 20CR and ERA-20C. The ordering of component pairs is not identical but there is a very clear correspondence of the spectral peaks. For instance, the 1.7 yr peak in the 20CR components 29–30 corresponds to the components 24–25 in ERA-20C. In summary, ordered according to the
  - explained variance. We can see that
    - the trend components with multi-decadal periods (components with predominantly multi-decadal periodicity (1and, 2) explain 6, 5, and 6) explain a total of 7.2% and 5.3-5.9% of the variance in 20CR and ERA-20C, respectively, with very similar spectra (the length of the time series 105 years restricts, of course, the correct identification of multi-decadal oscillations) the spectral peak at 3.5 yr and the broad one around 5 yr are very similar in components 3–6, although the associated 10–20 yr variability in-clear similarities in their time series and spectra
- 210

[revised manuscript text omitted]

removed absent from the datasets (, and therefore the red dots are far below the bars). The periods rising above the red noise at 5 and 10 significance level are tabulated in Table ??, columns 1 and 2, and are nearly the same fall far below their expected values. Interestingly, the period of 2.9 yr in 20CR and ERA-20C fall below the 95 % confidence interval. Our conclusion is therefore that the low-frequency-multi-annual climate variability in the near-surface air temperature is very similar in the two reanalyses20CR and ERA-20C.

**260 3.3 Evaluation of the simulated CMIP5 model total spectra**

The climate model simulation data was processed exactly the same way as the reanalysis data. The total spectrum of each model is shown in Figure ?? with total spectra for the 12 CMIP5 model are shown in Fig. ?? (solid lines) with their 95 % confidence intervals (dashed envelopes) and the reanalysis spectra on the background as a reference (dotted thin lines). The spectra are objective measures of model performance. We evaluate *subjectively* how the models reproduce Statistically

- significant multi-annual modes (at 5 % level) are denoted by vertical dashed lines. As in the case of reanalyses, these spectra are unique expressions of the low-frequency variability present in the simulation data. A comparison between the simulated and the reanalysis spectra , and adopt the following terminology: a model can be either under-active provides one means to assess the strengths and weaknesses of these models. However, one cannot simply rank the models based on how "far off" the model spectra are from the reference, because this comparison focuses on just one (although important) aspect of model
- 270 performance and because seemingly good agreement with observations might occasionally result from compensating errors in model processes.

Here we will concentrate on the multi-annual aspects but note in passing that the level of multi-decadal variability (over-active) if the spectral density is lower (higher) than in the reanalysis spectra, or a model is on-target with respect to spectral density. This enables us to make an overall evaluation of the current capabilities of climate models to represent low-frequency variability  $\div 20$  yr) is close to reanalyses in models a, c, d, e, and g. In the rest of the models, the level seems too low. In the decadal scale ( $\sim 10 \dots 20$  yr), the level of variance is close to reanalyses in a, b, c, f, i, j, and l. Subjectively, the shape of the low-frequency

end of the spectra appears most realistic in models a and c.

The subjective evaluation is summarised in Table ??. In representing multi-decadal variability, three models are on-target while the rest are under-active. In decadal variability, majority of the models are on target, while four are over-active. Only

280 one model has both of these variabilities on-target (a). In ENSO variability, two models (i and g) seem to have both the In multi-annual scales, the model performance varies a lot among the models. There is a group of models (a, b, d, and e) with high spectral density at about 3 – 7 yr periods. The models d and e have a bi-modal spectral structure, as in the reanalyses, while models a and b have a broad unimodal peak. Decompositions (available in the Supplementary material, S1) partly explain the reasons leading to these total spectra.

- 285 In model a, for instance, there is a unimodal broad peak at 3.5 and 5 yr periods on-target. Five models have one of these periods on-target, and the rest of the models are either under- or over-active. With respect to the inter-annual variability, four models seem to be over-active, while the rest are on target. 4 yr periods (Fig. ??a). The decomposition reveals that there are, in fact, two well separated component pairs at 3.5 yr and 4 yr generating one merged peak to the total spectrum (Fig. S1a in the Supplement). A development hint is thus to investigate these modes which can help to better understand some underlying
- 290 modelling deficiencies, and to keep monitoring how this aspect of model performance evolves in the future model upgrades. An additional concern in model a is the excessive spectral density at about 2 yr and 7 - 10 yr periods.

We are not going to rank the models, but two models (i and g) seem to perform particularly well and have four out of five key periodicities on-target. The IPSL-CM5B-LR model (i) is close to the reanalyses all the way from inter-annual and ENSO variability to the decadal scale of 20 years and only seem to have too little multi-decadal variability. The HadGEM2 model

- (gIn model e, there is a bimodal total spectrum (Fig. ??e), although far too pronounced as compared with the reanalyses. The decomposition (Fig. S1e in the Supplement) reveals that the ST-PC components 1 10 (except 7–8) are all multi-annual and peak strongly and well in isolation at 3 yr, 3.5 yr, 4 yr, and 5 yrs, explaining together no less than 13.9 % of the total variance. The development hint for model e is thus to investigate the mechanisms behind the components 1 10 and thereby obtain guidance for improving the realism of simulations.
- 300 In most other models, the multi-annual variability is less prominent than in the reanalyses. In model c (Fig. ??c), on the other hand, is reasonably close to the reanalyses across the spectrum, except for the notable over-activity at periods at and around 10 yr. CanESM2 (a)is the only model that closely reproduces both the decadal and one hand, the decomposition points out (Fig. S1c in the Supplement) that there are about 12 ST-PC components with periods between 1.5 3 yrs leading to a total spectrum with a broad peak of 2 3 yr periods. These components tend to have very regular cycles, remotely resembling a coupled
- 305 harmonic oscillator and seemingly missing the "offbeats" or true quasi-periodicity of the reanalyses. The task seems to be to find out reasons why model c produces too rapid and regular multi-annual variability. In model g (Fig. ??g), on the other hand, the leading ST-PC components 1 9 are on either decadal or multi-decadal variability periods and these overwhelm the total spectrum. It should be important to find out the causes for this accentuated variability, especially on the decadal scale.

Overall, the clearest signal here is that the modelsgenerally seem to lack multi-decadal variability , and some models are

310 over-active in representing decadal and inter-annual variability. In ENSO time scales, only two models are on-targetFinally, Fig. ?? casts light on models' overall level of variability compared to reanalyses. Clearly, this level in model h (Fig. ??h) is low. Curiously enough, the leading ST-PC component pair in model h explains only 1.4 % of variance and peaks at 3.2 yr. This corresponds to the isolated peak in the total spectrum.

**3.4 Significance test by Monte-Carlo MSSA of multi-annual modes in CMIP5 models**

315 Table ?? contains the periods rising In the reanalyses (Fig. ??), only a few multi-annual periods rise above the red noise at 5 and 10 significance level. The periods for (three in 20CR and four in ERA-20Care in the columns 1 and 2. Both data sets have five periodicities significant at ). They are at approximately 3.5 yr and 5 level, four of which are common and one specific. Additionally, there are three common periodicities significant at 10 level. In terms of statistical significance, 20CR and ERA-20C behave in a very similar manner.

- 320 Table ?? gives an impression that the number of statistically significant periodicities is very large in models, in general. In this respect, the climate model data is dissimilar to the reanalysis data. There is only one model (model k) which has fewer significant periodicities than the reanalyses, and three models (g, h, j) are somewhat close to reanalyses with 11–13 significant periodicities. The rest of the models have many more periodicities (up to 30). Interestingly, yr periods. For the CMIP5 models, the test results are available in the Supplementary material (S2). In Fig. ??, the multi-annual modes with periods less than 7
- 325 yrs at the HadGEM2 model (g) has the 10 yr peak, discussed earlier, significant at 5 % level.

The number of models that correctly detected either the 20CR or ERA-20C periodicities were seven models for significance level are denoted by dashed vertical lines.

In summary, there are 5 - 15 statistically significant periods in the models, except model k (Fig. **??**k) with three and model g (Fig. **??**g) with zero periods. The large number of significant periods (models d and e, for instance) can be explained, at least

- partly by the fact that the modes are quasi-periodic and the spectral density therefore appears on a range of frequencies. This manifests as excursion of the 1.7 yr period, five for 2.2 yr, four for red-noise threshold on several adjacent frequencies. This is typical for models with large spectral power on certain time scales. In model 1 (Fig. ??!), for instance, there are two broad and distinct spectral peaks at about 3.5 yr, seven for 3.6, four for 4.2 yr, and two for 5.2 and 6 yr periods, and many significant periods are gathered at these and nearby frequencies. In contrast, models f and h (and to some extent model c) have several significant and distinct periods between 2 yr and 7 yr. In addition to these, there were many "false alarms", as seen in Table
- 335 significant and distinct periods between 2 yr and 7 yr. In addition to these, there were many faise analysis, as seen in fable ??terms of number of significant modes, models a, i, j, and k seem to be closest to the reanalyses.

**3.5 Spatial patterns of the 3.5 3-4 yr mode**

340

The oscillatory mode with a 3.5 yr period was significant in 20CR and ERA-20C and in seven climate models. The ST-PC components can be represented in the original coordinate system as so called reconstructed components (RC) that can be visualised. In this section, some visualisation results are presented and discussed.

In ERA-20C, there is a spectral peak at 3.5 yr has some power on both sides of the peak. Therefore, a 3–4 yr mode would perhaps be a more appropriate name for the peak. Oscillations at these periods are usually attributed to ENSO-period, which is significant at 5 % level (Fig. ??). This peak is due to the ST-PC components 7 and 8 with spectral density closely concentrated on this frequency (Fig. ??). Figure ?? depicts composite maps of each of the eight phases of the 3.5 yr mode in ERA-20C.

345 Firstly, the mode is global with the largest temperature anomalies in the Pacific and North-America. Secondly, the mode contains tropical Pacific temperature anomalies, like in the ENSO phenomenon (e.g. Kleeman, 2008). We illustrate here how this periodicity appears in different data sets. Since the 3.5 yr period is not reproduced by all of the climate models, we have chosen in these cases a periodicity that is close to 3.5 years.

We have calculated the reconstructed components (RC) corresponding to the 3.5 yr mode in order to visualise the associated
 spatial temperature anomaly patterns. The result is a time series of the original (centered)data corresponding to the 3.5 yr mode in each grid point and according to its original variance. We therefore have an animation of The cold (warm) maximum is in

phase 1 (5) with the anomalies extending to the global 3.5 yr mode for each data set for the whole timeperiod (1901–2005). South-American continent. Thirdly, there are traveling temperature anomalies at high latitudes on both hemispheres. These are described next.

- **355** To synthesise the animations, we have calculated composite maps of the 3.5 yr mode for each data set In phase 1 (Fig. ??), there is a small warm temperature anomaly in the North-Pacific (lon 160°W, lat 30°N). This pattern slowly moves northeast reaching Alaska in phase 5 and then gradually dissipating over the northernmost North-America in phase 8 (and being visible still in phase 1). There is a very similar evolution of a cold anomaly starting in phase 5. At the same time, there is an oscillating temperature anomaly over the Eurasian continent in opposite phase. In Fig. ??, there is also a traveling temperature anomaly
- 360 in the Southern Hemisphere. In phase 8 (Fig. ??). The patterns are composites of eight instances when the mode is in its maximum positive phase in the Niño3.4 region (120??), there is a cold anomaly over the Southern Ocean (lon 160°W–170W). This strengthens, moves east, weakens, and crosses the Antarctic Peninsula in phase 4 and remains in the Weddell Sea until phase 7. Similarly, there is a warm anomaly in phase 4 (lon 160°W, 5° N–5° S). Positive events are defined as an average of winter months (November–March) with similar evolution as the cold one.
- 365 The top row of Fig. ?? displays the patterns for the reanalyses. One can see typical El Niño related temperature anomalies, such as positive anomalies in the equatorial Pacific Ocean and South-America, Next, 20CR and the CMIP5 model behaviour is studied. The 3.5 yr mode is significant in 20CR and northwestern North-America. There is also a cold anomaly over the northern parts of the Eurasian continent, a dipole structure in the western Antarctica, and warm Africa. The patterns are remarkably similar ERA-20C. For the illustration, we have chosen component pairs from the model decompositions (Supplementary
- 370 material Fig. S1) that have spectral peaks between 3 and 4 years and do not express substantial variability on other time scales. In most climate models, such a corresponding mode exists, except in models g and k. In model c this mode is not significant at 5 % level, but it is illustrated anyway. Supplementary material reveals how these modes are represented in different data sets (Fig. S3–S14). The format is the same as in Fig. ??. A short summary is presented next.
- In 20CR (Fig. S3), the anomalies are weaker compared to ERA-20C (Fig. S4). This is mainly because the 3 4 yr mode is distributed on two component pairs in 20CR and whereas in ERA-20C with somewhat larger amplitudes it is concentrated on one pair. Nevertheless, similar although weaker signal is evident in 20CR, except the stronger dipole in ERA-20C. Next, we will try to *subjectively* assess model performance in reproducing the spatial patterns of such as the northeast propagation of the North-Pacific temperature anomaly. (Note that in Fig. ??, the 3.5 yr oscillation.
- All models produce a warm pool combination of components 3, 4, 7, and 8 produce highly similar global patterns for 20CR
   and ERA-20C.) A prominent feature is also the opposite temperature anomalies in the northern Eurasia versus North-America.
   All models (Figs. S5–S14) produce a temperature anomaly to the equatorial Pacific Ocean (and South-America). The amplitude is larger and/or the area extends too far further to the west in five than in ERA-20C in six models (a, b, d, e, h). Unlike in the reanalyses, the pool extends to the Atlantic in four models (a, d, e, h). The warm anomaly in the South-American land area is too weak in five models (e, f, g, j, k).

385 The warm anomaly, 1). The anomaly pattern in the northwestern North-America is present in all the models to some extent. In the reanalyses, the anomaly is strictly confined to land areas but in most models, it is either somewhat misplaced or extends to the adjacent sea areas and the Eurasian continent.

The cold anomaly over the Eurasian continent is not well represented in the model data. There is a weak cold anomaly in three models (f, i, l), a weak warm anomaly in one model (c), and a mixture of cold and warm areas in the rest.

- 390 The warm pool in the Amundsen sea related to the warm-cold dipole around the western Antarctica is present in nine models (aModels c, e, f, g, h, i, j, k, l). The cold pool in the Weddell sea is present in five models (c, d, f, g, k). Three models (f, g, k) represent both pools, and thus the dipole structure in well represented. In Africa, the anomaly is on the positive side in all models.
- In Table ??, there are three over-active models with respect to and f produce the North-American pattern quite similar to 395 reanalyses, and the 3.5 yr period (a, d, c). These are associated with large amplitudes in Fig. ??. Additionally, two models also seem to have large amplitudes in Fig. ?? (b, h). Both of these have higher spectral density than the reanalyses (Table ??), but were anyhow assessed as "on-target" in Table ??. Four models were under-active in Table ?? (northeast propagation is captured to some extent by models b, c, f, j, k). These correspond to the low-amplitude maps in Fig. ??. In summary, the patterns of Fig. ?? are in support of the subjective analysis of the total spectra of Table ?? i, and 1.

**400 4 Discussion**

Table ?? shows that there is a much larger number of significant periodicities in the model data than in the reanalyses. This seems to imply that "nature" tends to produce only a few but statistically significant periodicities, and the other potential periodicities are somehow. We note that a substantial portion of variance at inter-annual to inter-decadal timescales can be attributed to "worn outclimate noise" via non-liner interactions and feedbacks so that they cannot be distinguished from red

- 405 noise. We can think of two possible causes for this. First, it may simply be because the reanalysis data represent ensemble mean while the model data are individual simulations. This difference may appear as a difference in the number of significant periodicities. Second, it may be that something in models prohibits the wearing process from occurring, and we can observe the excessive number of periodicities above the noise level as if the models were missing some non-linear processes. However, MPI-ESM (model k)has fewer significant periodicities than observed, and to our knowledge, this model does not fundamentally
- 410 differ from the other models. Therefore, it is unclear what exactly lies behind this result.

We have not discussed the interpretation of the 1.7 yr interannual variability. Based on the related spatial patterns, it may be that it is a harmonic of the ENSO variability  $(2 \times 1.7 \text{ yr} = 3.4 \text{ yr})$  associated with processes with timescales much shorter than the inter-annual scale (Wunsch 1999; Feldstein 2000). If the amplitude of the variability mode exceeds some noise threshold (such as red noise), then the variability mode is also likely driven by some process external to the atmosphere, in addition to

415 the climate noise. For example, large part of the inter-annual atmospheric ENSO pattern is presumably driven by anomalies of tropical diabatic heating associated with sea surface temperature anomalies (Feldstein, 2000). We assume that for this reason the multi-annual patterns related to ENSO clearly exceed the noise threshold in the results of this study.

**5** Conclusions**

In this study, we decomposed The aim of this study is to decompose the 20th Century near-surface temperature century

- 420 climate variability into its inter-annual to multi-decadal eigenmodes. We used two state-of-the-art-multi-annual modes, and to assess how these modes are represented by the contemporary climate models. To this end, two 20th Century century reanalysis data sets and 12 historical climate model simulations extracted from the CMIP5 data archive. The analysis was performed using the randomised multi-channel singular spectrum analysis CMIP5 model simulations for years 1901–2005 of the monthly mean near-surface air temperature have been decomposed using Randomised Multi-Channel Singular Spectrum
- 425 Analysis (RMSSA), which is particularly well suited to high-dimensional time series analysis. The statistical significance of the identified eigenmodes was estimated with Monte-Carlo modes has been estimated with Monte Carlo simulations. The main conclusions are as follows:

The spectral Spectral properties of the two reanalyses (20CR and ERA-20C) are remarkably similar, the only notable difference being the different spectral density in the decadal scale variability (10–30 yr). Also, nearly the same periodicities

- 430 rise above the red noise at the 5 and 10 significance levels. Majority of the climate models are under-active in representing the multi-decadal climate variability (> 30 yr), some models are over-active in decadal (10–30 yr) or inter-annual (< 2 yr) variability, and only two models are on-target in both the reanalysis data appear remarkably similar. The most prominent forms of variability in both data sets are related to approximately 3.5 yr and 5 yr ENSO variabilities. The IPSL-CM5B-LR and the HadGEM2-ES are the closest to the total spectra of yr modes which are significant at 5 % level. The spectral power in ERA-20C
- 435 is systematically slightly higher than in 20CR. The 3.5 yr mode is illustrated in more detail. In ERA-20C, the reanalyses. mode is associated with typical ENSO pattern of temperature anomalies in the equatorial Pacific Ocean, South-America, and northwestern North-America. On top of these, the mode also contains a northeast propagating temperature anomaly over the northernmost North-America, and another eastward propagating anomaly in the vicinity of western Antarctica. Since about the 1970's, the amplitude of this 3.5 yr global mode have increased.
- 440 Relaxation None of the 12 coupled climate models closely reproduce all aspects of the reanalysis spectra, although some models represent many aspects well. For instance, the GFDL-ESM2M model has two nicely separated ENSO -related periods although they are relatively too prominent as compared to the reanalyses. Also, a number of models represent the propagating temperature anomalies at 3 4 yr time frame. Some suggestions are provided in the text for potential model development aspects.
- 445 There is an extensive Supplement available presenting the results in visual format for each reanalysis and model data set. In the future, relaxation of the uni-variate nature of the present study remains as a subject for future research. It would be very interesting to extend the set of variables 
[revised manuscript text omitted]

| 1        | MRI-CGCM3     | MRI/JMA           | Japan     |

Evaluation summary. Under-active On-target Over-active Multi-decadal (>30 yr) b, c, d, e, f a, g, j h, i, k, l Decadal (10-30 yr) a, b, c, e, f,

d, g, j, k h, i, 1 ENSO at ~5 yr a, f, h, j, 1 c, g, i, k b, d, e ENSO at ~3.5 yr c, f, j, k b, g, h, i, 1 a, d, e Inter-annual (